# An organic/inorganic electrode-based hydronium-ion battery

Zhaowei Guo[1], Jianhang Huang[1], Xiaoli Dong [1], Yongyao Xia[1], Lei Yan[1], Zhuo Wang[1] & Yonggang Wang [1⊠]

Hydronium-ion batteries are regarded as one of the most promising energy technologies as next-generation power sources, benefiting from their cost effectivity and sustainability merits. Herein, we propose a hydronium-ion battery which is based on an organic pyrene-4,5,9,10-tetraone anode and an inorganic $MnO_2$@graphite felt cathode in an acid electrolyte. Its operation involves a quinone/hydroquinone redox reaction on anode and a $MnO_2/Mn^{2+}$ conversion reaction on cathode, in parallel with the transfer of $H_3O^+$ between two electrodes. The distinct operation mechanism affords this hydronium-ion battery an energy density up to 132.6 Wh kg$^{-1}$ and a supercapacitor-comparable power density of 30.8 kW kg$^{-1}$, along with a long-term cycling life over 5000 cycles. Furthermore, surprisingly, this hydronium-ion battery works well even with a frozen electrolyte under −40 °C, and superior rate performance and cycle stability remain at −70 °C.

[1] Department of Chemistry and Shanghai Key Laboratory of Molecular Catalysis and Innovative Materials, Institute of New Energy, iChEM (Collaborative Innovation Centre of Chemistry for Energy Materials), Fudan University, Shanghai 200433, China. ⊠email: ygwang@fudan.edu.cn

Lithium ion batteries (LIBs) using organic electrolytes have been extensively used to power various portable electronics and electric vehicles (EVs) because of their desirable electrochemical performance[1–9]. Subsequently, the LIBs employing aqueous solution electrolytes (i.e., aqueous LIBs) were introduced to relieve safety hazard associated with combustible and toxic organic electrolytes[10–16], enabling the implementability of derived high-safety applications, such as wearable electronic devices and large-scale energy storage. Nonetheless, the limited natural resource of lithium mineral still handicaps the further wide applications of LIBs, which is confirmed by the recently sharp increase of Li salts price. As a result, the earth-abundant metal ions, such as $Na^+$, $K^+$, $Mg^{2+}$, $Zn^{2+}$ and $Al^{3+}$, are being considered as the alternative charge carriers for building rechargeable hydronium-ion batteries. Unfortunately, owing to the higher ionic radius and/or charge number than $Li^+$, these batteries generally show the inferior performance to LIBs[17–30].

Besides these metal ions, proton ($H^+$) can also be used as a promising charge carrier due to its smallest size and lightest weight among all cation ions on the earth[1,31–37]. However, the naked $H^+$ can rarely be used as charge carrier because of the formation of hydronium ions ($H_3O^+$), where the high dehydration energy of $H_3O^+$ (11.66 eV) prevents the de-solvent process ($H_3O^+ \rightarrow H^+ + H_2O$). That is to say, $H_3O^+$ can generally be used as the charge carrier, rather than the naked $H^+$. The hydronium ($H_3O^+$) charge carrier shows the cost effectivity and sustainability advantages over these metal ions. Unfortunately, only several hydronium-ion batteries have been demonstrated recently, which might be attributable to that the size of $H_3O^+$ (that is larger than naked $Li^+$ and close to naked $Na^+$) limits choice of host materials. Very recently, Ji et al. demonstrated that some host materials could reversibly store proton-based charge carrier (i.e. $H_3O^+$ or $NH_4^+$), which evokes the enthusiasm for building hydronium-ion batteries[35–39]. However, the achieved performances are still inferior to previous aqueous LIB batteries, showing the limited capacity ($\leq$130 mAh g$^{-1}$) or cycle stability (<1000 cycles). Therefore, it is necessary to explore new battery chemistry for $H_3O^+$ storage.

Herein, we propose a hydronium-ion battery, in which $H_3O^+$ is used as charge carrier to couple the reversible quinone/hydroquinone redox reaction in a pyrene-4,5,9,10-tetraone (PTO)-based anode and the reversible $Mn^{2+}/MnO_2$ conversion reaction in a carbon-based cathode. Figure 1 illustrates the structure of the hydronium-ion battery, which involves a PTO-based anode and a graphite felt (GF) electrode (that serves as both substrate and current collector for cathode) in an acid electrolyte containing $Mn^{2+}$ (i.e., 2 M $MnSO_4$ + 2 M $H_2SO_4$). On charge, the $Mn^{2+}$ in electrolyte is oxidized into $MnO_2$ precipitate on GF electrode with generation of $H_3O^+$ (Eq. 1), and simultaneously $H_3O^+$ in the electrolyte is stored by PTO to form hydroquinone (HQ) through the reaction shown in Eq. 2. Electrons obviously transfer from cathode to anode through external circuit. The discharge reverses the charge process. The overall charge/discharge operation mechanism is as follows:

$$Mn^{2+} + 6H_2O \leftrightarrow MnO_2 + 4H_3O^+ + 2e^- \quad (1)$$

$$PTO + 2e^- + 2H_3O^+ \leftrightarrow PTO - 2H + 2H_2O \quad (2)$$

The hydronium-ion battery has been demonstrated to exhibit a high energy density and a supercapacitor-like power density, along with a long-term cycle life over 5000 cycles. Furthermore, the hydronium-ion battery in a flexible configuration has also been built for flexible and wearable electronic devices, showing high safety and robust mechanical property. In addition, surprisingly, we find that this hydronium-ion battery is able to exhibit promising electrochemical performance at the ultra-low temperature

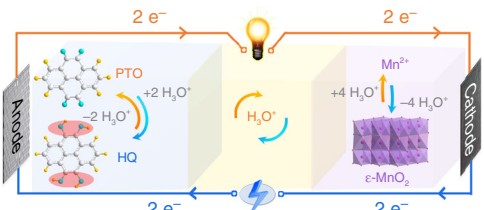

**Fig. 1** Schematic illustration of working mechanism for the PTO//MnO$_2$@GF hydronium-ion battery.

($-40$ to $-70$ °C) in spite of the freeze of electrolyte at such temperature, which provides an approach for developing low-temperature batteries.

## Results

**Electrochemical properties of half-cells.** Prior to the fabrication of full battery, the electrochemical properties of anode (i.e., PTO-electrode) and cathode (i.e., graphite felt-electrode) in the acid electrolyte containing $Mn^{2+}$ were examined by the typical three-electrode tests, respectively. PTO sample was prepared according to our previous report[28]. Commercialized graphite felt electrode was directly used for the tests without further treatment. The as-prepared PTO particles display a well-ordered morphology in the shape of parallel oriented nanorod arrays with a diameter of 200–500 nm and a length of 1–5 μm (Supplementary Fig. 1). The sharp diffraction peaks of X-ray diffraction (XRD) patterns in Supplementary Fig. 2 reveal a fine crystallinity of PTO, and its high purity is verified by $^1$H-NMR (Supplementary Fig. 3) and FT-IR spectra (Supplementary Fig. 4).

Figure 2a presents the cyclic voltammetry (CV) curves of PTO-electrode tested at different scan rates, where the electrode exhibits two pairs of symmetric peaks, showing a highly reversible two-step discharge/charge process. Furthermore, the peak currents increase almost linearly with the increase of scan rates (see Supplementary Fig. 5 and corresponding discussions). The galvanostatic discharge-charge profiles with two pairs of well-defined sloping plateaus in Fig. 2b could be detected in accordance with above CV results. The PTO-electrode delivers a reversible specific capacity of 208 mAh g$^{-1}$ at a current density of 0.16 mA cm$^{-2}$ (corresponding to 0.2 C, defining 1 C as 0.8 mA cm$^{-2}$), and a decent capacity of 85 mAh g$^{-1}$ is delivered even at an extreme high current density of 480 mA cm$^{-2}$ (600 C), proving an ultrafast reaction kinetics and good rate ability (Fig. 2b). It should be noted that the achieved capacity (208 mAh g$^{-1}$) at a low current density is only about half of the theoretical capacity of approximately 400 mAh g$^{-1}$ (calculated based on 4e4H reaction), showing that only ~50% carbonyl groups can be used for $H_3O^+$ storage. This phenomenon is further confirmed by the FT-IR characterization later. Owing to the insolubility and structural stability during cycling, a high capacity retention of 78% (corresponding to a specific capacity of 118 mAh g$^{-1}$) is still attainable over 1000 cycles at a moderate current density of 2 mA cm$^{-2}$ (2.5 C) (Fig. 2c). In addition, the hybrid electrolyte (2 M $MnSO_4$ + 2 M $H_2SO_4$) contains both $H_3O^+$ and $Mn^{2+}$. Therefore, it is necessary to clarify which one ($H_3O^+$ or $Mn^{2+}$) plays the role of charge carrier for reaction on PTO-electrode. In order to reveal the intrinsic mechanism clearly, a series of ex-situ characterizations have been carried out to investigate the electrode-active material (i.e., PTO) evolution at various discharged/charged states. The acquired STEM EDS-elemental mapping images (Fig. 2d) and XPS spectra (Supplementary Fig. 6) of pristine, fully-reduced and fully-recovered electrodes clearly indicate that $Mn^{2+}$ is not the charge carrier, on account of the absence of Mn element on PTO-electrode at the fully-reduced state. Accordingly, it can be assumed that $H_3O^+$ is

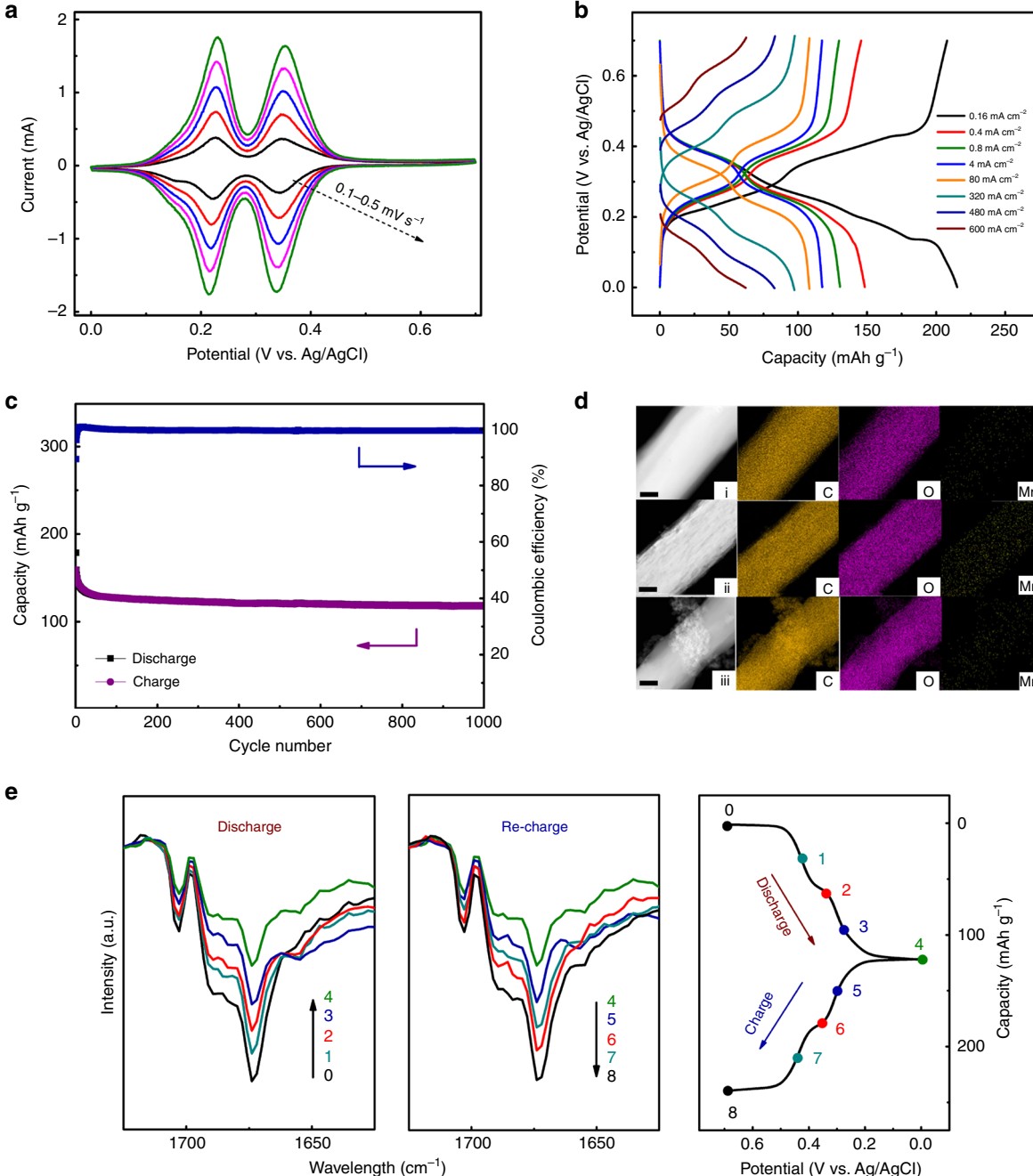

**Fig. 2 Demonstrations of the H$_3$O$^+$ release/capture mechanism on PTO-electrode. a** Cyclic voltammetry (CV) curves of PTO-electrode at various scan rates from 0.1 to 0.5 mV s$^{-1}$. **b** Galvanostatic discharge-charge profiles of PTO-electrode at different current densities. **c** Cycle performance of PTO-electrode at a current density of 2 mA cm$^{-2}$. **d** TEM-HADDF and STEM EDS-elemental mapping images of PTO-electrodes at (i) pristine, (ii) fully-reduced, and (iii) recovered states. Scale bar: 100 nm. **e** Ex-situ FT-IR spectra of PTO-electrode at various discharged/charged states during a galvanostatic cycle (as marked points on the right discharge-charge curves). All the electrochemical tests for characterization were conducted at a current density of 2 mA cm$^{-2}$ (2.5 C) in a three-electrode system. 2 M MnSO$_4$ + 2 M H$_2$SO$_4$ solution were used as electrolyte.

involved in the reaction on PTO-electrode (as shown in Eq. 2) as charge carrier. To verify this point, electrochemical behaviors of PTO-electrode were probed in a single Mn$^{2+}$-containing (i.e. 2 M MnSO$_4$) electrolyte and a single H$_3$O$^+$-containing (i.e., 1 M H$_2$SO$_4$) electrolyte for comparison. As is shown in Supplementary Fig. 7, the electrochemical profile of the PTO-electrode in the hybrid electrolyte (2 M MnSO$_4$ + 2 M H$_2$SO$_4$) is as the same as that in a single H$_3$O$^+$-containing (2 M H$_2$SO$_4$) electrolyte, which confirms the above assumption. The active sites on PTO molecules were also examined by ex-situ FT-IR spectra at various discharged/

charged states as marked in the right of Fig. 2e. The characteristic absorption peaks at ~1670 cm$^{-1}$ is ascribed to carbonyl groups of PTO molecule, whose intensity is positively correlated with the content of pristine carbonyl groups. During the reduction (discharge) process, the reduction depth of active groups on PTO increases along with the potential decrease. While during the oxidation (re-charge) process, the oxidation depth of active groups increases with the potential increase. Therefore, the content of redox-engaged carbonyl groups reflected by absorption peak intensity will evolve with discharge/charge potential. As shown

in Fig. 2e, the peak intensity of at ~1670 cm$^{-1}$ decreases with the reduction depth increase of PTO-electrode. Subsequently, these peaks strengthen back gradually to approach that of pristine state during re-charge process (Fig. 2e), confirming the reversible quinone/hydroquinone redox reaction occurring on PTO-electrode (see Supplementary Fig. 8 for the full FT-IR spectra). It should be noted that the FT-IR spectra shown in Fig. 2e and Supplementary Fig. 8 were manipulated by a routine baseline correction and background removal process for the clear observation of characteristic absorptions. In addition, the FT-IR spectra performed at different states of oxidation show that even in the fully reduced form there is absorption corresponding to carbonyl vibration remaining, indicating that not all of carbonyl groups can be used for charge storage (that consists with charge/discharge results).

Besides investigations about electrochemical profile of PTO-electrode, the deposition/dissolution behavior of $MnO_2$ on GF electrode in the hybrid electrolyte (2 M $MnSO_4$ + 2 M $H_2SO_4$) was also characterized by three-electrode tests. As is shown in Fig. 3a, a pair of redox peaks at around 1.18 V (vs. Ag/AgCl) present in the CV curves at a scan rate of 0.1 mV s$^{-1}$, which should be assigned to the reversible conversion of $Mn^{2+}$/$MnO_2$ coupled with the generation/consumption of $H_3O^+$ (Eq. 1). Voltammetric responses at various scan rates (i.e. peak currents vs. scan rates) indicates the reversible deposition/dissolution reaction on GF electrode (see Supplementary Fig. 9 and corresponding discussions). Galvanostatic charge-discharge profiles were acquired at different current densities (mA cm$^{-2}$) with a fixed $MnO_2$ deposition capacity (i.e., 1 mAh) as Fig. 3b shows, presenting one redox potential plateau in accordance with CV results. The polarization gradually increases with the increase of applied current density from 0.5 to 100 mA cm$^{-2}$ (Fig. 3b), with the coulombic efficiency of 95% at a low current density of 0.5 mA cm$^{-2}$. SEM images shown in Supplementary Fig. 10 indicate that the GF electrode is composed with a lot of carbon fibers (i.e., GF fibers). Morphologies of these GF fibers before/after depositing $MnO_2$ were examined by ex-situ SEM characterization (Fig. 3c and Supplementary Fig. 11). It can be observed from Fig. 3c and Supplementary Fig. 11 (that is the magnification of Fig. 3c) that flake-like $MnO_2$ nanocrystalline gathers gradually to grow into numerous individual urchin-like clusters on the smooth surface of GF fibers, and then forms a compact interconnected-nanoflakes $MnO_2$ layer covering the surface of GF fibers entirely. However, the $MnO_2$ layer begins to exfoliate under subsequent electrodeposition due to excessively thick of $MnO_2$ layer when the deposition capacity reaches 10 mAh (see Supplementary Fig. 12 and discussion in Supplementary Note 1). The SEM EDS-elemental mapping images of GF electrode after $MnO_2$ deposition with different applied currents and capacities are given in Fig. 3d and Supplementary Fig. 13, indicating the uniformly distributed $MnO_2$ layer on GF surface. The XPS spectra of the GF electrode after electrodeposition process reveals the charge state of Mn as ~4.0 and the derived quantified Mn/O atomic ratio of 0.5, namely $MnO_2$ molecular formula[40] (Fig. 3e and Supplementary Fig. 14). The electrodeposited $MnO_2$ particles are indexed to Akhtenskite phase (JCPDS 30-0820) from characteristic peaks of XRD patterns, where peak intensities increase with $MnO_2$ electrodeposition accordingly, except for the peaks at 26° ascribed to GF substrate (Fig. 3f). The weak-strength and broad peak characteristic indicates the electrodeposited $MnO_2$ particles are nanocrystalline. The deposited $MnO_2$ dissolves into electrolyte on recharge process, which is clarified by Supplementary Fig. 15. It is well demonstrated that discharge of $MnO_2$ in acid includes both the ion intercalation (electrochemical process) and the disproportionated reaction of $Mn^{3+}$ (chemical process), through which $MnO_2$ is converted into $Mn^{2+}$ (see the discussion in Supplementary Note 2 for details)[41–44].

**Electrochemical performance of the hydronium-ion full battery**. After electrodes investigation, the full batteries were fabricated according to Fig. 1, in which PTO-electrode and the GF-electrode pre-deposited with some $MnO_2$ (that is $MnO_2$@GF electrode) were used as anode and cathode, respectively, with a hybrid electrolyte (2 M $MnSO_4$ + 2 M $H_2SO_4$). Herein, the pre-deposition treatment of $MnO_2$ was just used to compensate the slight difference of Coulombic efficiencies between anode and cathode on initial cycles (see methods section). The galvanostatic charge/discharge profile (voltage vs. time) of the full battery and the potential responses (potential vs. time) of individual electrodes are given in Fig. 4a. It can be detected that the battery displays a slope voltage profile within the voltage window 0.3–1.3 V, which arises from the potential difference between cathode (approximately 1.15 V vs. Ag/AgCl) and anode (0–0.7 V vs. Ag/AgCl). Rate performance (voltage vs. capacity at different applied current densities) of the battery is given in Fig. 4b. It delivers a high capacity of 210 mAh g$^{-1}$ at a low current density of 0.16 mA cm$^{-2}$ (0.2 C), calculated based on the active material mass of PTO-electrode. Even at an extremely high current density of 400 mA cm$^{-2}$ (500 C), a respectable capacity of 66 mAh g$^{-1}$ is achieved (Fig. 4b), indicating an ultrafast redox reaction. It should be noted that one cell was used for various rates examination (Fig. 4b), thus partial capacity reduction at higher rates was due to the slightly capacity decay on cycling. Derived from the data shown in Fig. 4b, the full battery delivers a maximum energy density of 132.6 Wh kg$^{-1}$ at a power density of 25.6 W kg$^{-1}$, calculated based on the total mass of PTO and deposited (or dissolved) $MnO_2$ on cycling (see Supplementary Note 3 for detailed calculations). The maximum power density of 30.8 kW kg$^{-1}$ is achieved with a decent energy density of 20.0 Wh kg$^{-1}$. As shown in Supplementary Table 1, the achieved performances are superior to state-of-the-art proton-based batteries/capacitors[31–35]. It should be noted that the intercalation reaction-based batteries, such as Li-ion batteries and Na-ion batteries, generally exhibit very limited self-discharge, because the charge carriers (Li$^+$ or Na$^+$) are stored in the framework of electrode materials. Does the PTO// $MnO_2$@GF hydronium-ion battery has the similar merit (i.e., low self-discharge)? To clarify this point, a self-discharge test was conducted by programming the battery to be fully-charged firstly, then undergoing fully-discharged process after a 24 h rest. It can be observed from Fig. 4c that the battery voltage retains stable at around 0.94 V during the whole rest process, and the delivered discharge capacity (150 mAh g$^{-1}$) is approximately to charge capacity (152 mAh g$^{-1}$). Furthermore, the full battery performs a long-term cycle stability over 5000 cycles at a current density of 2 mA cm$^{-2}$ (2.5 C), with most of capacity decay in the first 50 cycles. The capacity loss is only 20% from 50$^{th}$ to 5000$^{th}$ cycles (Fig. 4d), exhibiting a quite robust and stable cyclability performance. In above investigation of full battery, the charge/discharge depth of cathode depends on the mass loading of PTO in cathode. For example, when cycled at a low current of 0.16 mA cm$^{-2}$, the deposited $MnO_2$ is 1.33 mg cm$^{-2}$ that is calculated based on the mass loading (4 mg cm$^{-2}$) and corresponding capacity (208 mAh g$^{-1}$) of PTO. Furthermore, with much higher mass loading of PTO (25 mg cm$^{-2}$), the deposited $MnO_2$ on cycling can reach 8 mg cm$^{-2}$ (see Supplementary Fig. 16 and related discussion in Supplementary Note 4 for details). These achievements indicate the potential application of PTO//$MnO_2$@GF hydronium-ion battery for grid-scale energy storage. To demonstrate the application prospect for wearable electronic devices, a flexible belt-shaped PTO//$MnO_2$@GF hydronium-ion battery in "sandwich"-configuration has been fabricated (Supplementary Fig. 17). The flexible battery exhibits a remarkable rate ability as expected, along with high volumetric energy and power densities, much higher than most of the previously proposed flexible batteries and

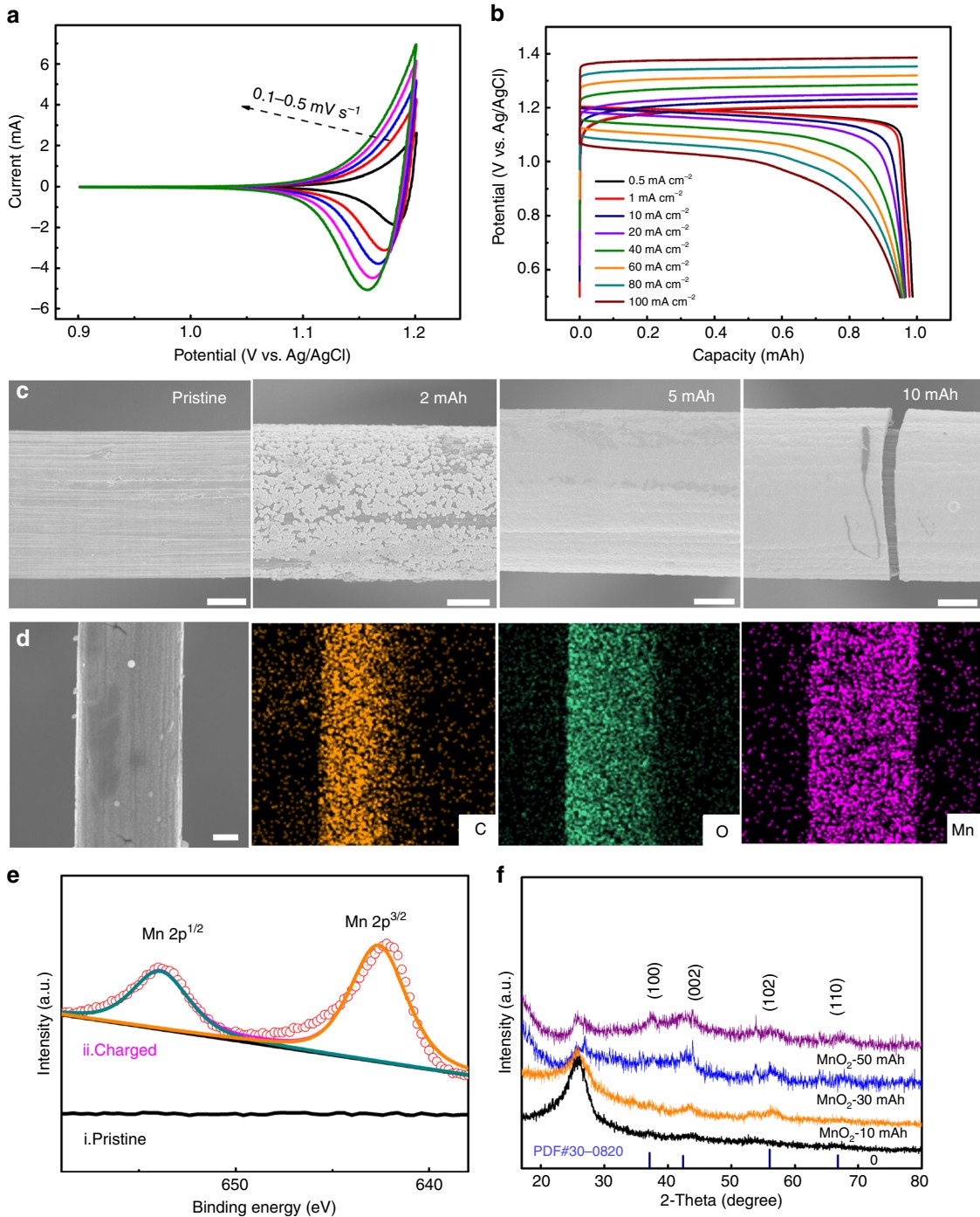

**Fig. 3 Electrochemical performance and morphology characterization investigations of MnO₂@GF cathode. a** CV curves of GF-electrode at various scan rates from 0.1 to 0.5 mV s⁻¹. **b** Galvanostatic discharge-charge profiles of GF-electrode at different current densities. **c** Morphology characterizations by SEM at pristine, 2 mAh, 5 mAh, and 10 mAh MnO₂ electrodeposition situations. Scale bar: 2 μm. **d** SEM images and EDS-elemental mapping images with 5 mAh MnO₂ electrodeposition for GF-electrode. Scale bar: 2 μm. **e** XPS spectra of Mn element on GF-electrode at (i) pristine state, and (ii) charged state with 5 mAh electrodeposition capacity using a charge current density of 5 mA cm⁻². **f** XRD patterns for MnO₂@GF cathode under various MnO₂ electrodeposition capacities (i.e., 0, 10 mAh, 30 mAh, 50 mAh, with an applied current density of 20 mA cm⁻²). All above tests used 2 M MnSO₄ + 2 M H₂SO₄ solutions as electrolytes.

supercapacitors (see Supplementary Fig. 18 and Supplementary Table 2 for detailed comparison). Moreover, the belt-shaped battery achieves an extended cycling stability over 1000 cycles, robust mechanical property and bending-resistant ability (see Supplementary Fig. 18 and Supplementary Note 5).

**The low-temperature performance of the hydronium-ion battery.** Our previous reports have demonstrated that the electrode materials relying on intercalation mechanism undergo sluggish desolvation process of charge carrier ions at low temperature, which leads to poor low-temperature behavior[45]. As mentioned

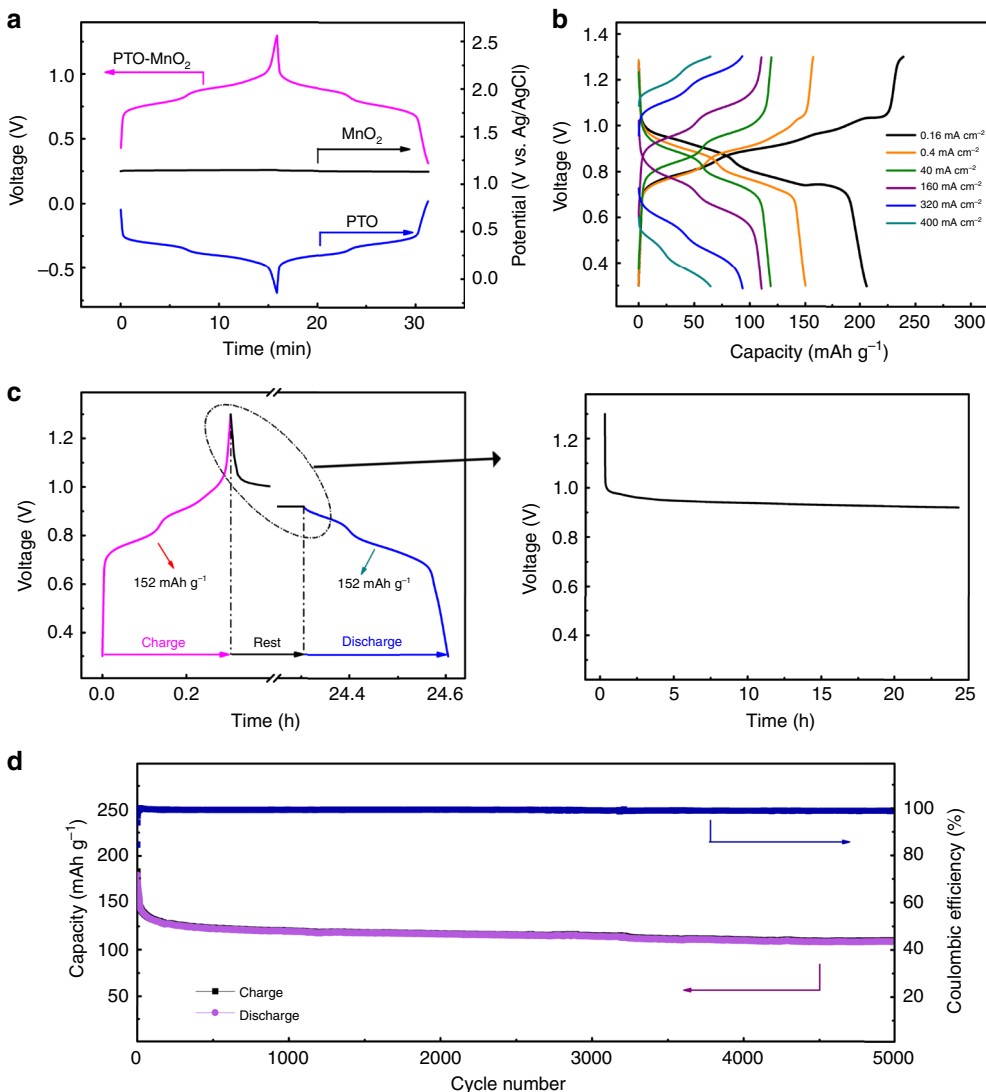

**Fig. 4 Electrochemical performances and stability property study of the PTO//MnO$_2$@GF prototype battery using [2 M MnSO$_4$ + 2 M H$_2$SO$_4$] electrolyte. a** Galvanostatic charge-discharge profiles of PTO//MnO$_2$@GF full battery and synchronic typical individual charge-discharge profiles of PTO-anode and MnO$_2$@GF-cathode, respectively, tested at a current density of 2 mA cm$^{-2}$ (2.5 C). **b** Rate ability described by galvanostatic charge-discharge curves at various current densities. **c** Electrochemical stability of the full battery surveyed by self-discharge experiments. Specifically, the full battery was fully charged to 1.3 V, then fully discharged to 0 V after rest for 24 h. **d** Cycle stability of the full battery tested at a current density of 2 mA cm$^{-2}$ (2.5 C).

above, the hydronium-ion battery does not include interaction reaction in electrodes. In addition, H$_3$O$^+$ can be directly combined with carbonyl groups of PTO without desolvation process. Therefore, it might be able to work well at low temperature. To clarify this point, we investigated the electrochemical behavior of the PTO//MnO$_2$@GF hydronium-ion battery at various low temperatures ($-40$ °C to $-70$ °C). Prior to these experiments, the freezing point of this hybrid electrolyte (2 M MnSO$_4$ + 2 M H$_2$SO$_4$) was examined. Such liquid electrolyte has been observed to transform into solid-state under $-40$ °C (Fig. 5a), indicating that the solid-state hydronium-ion battery can be formed when operated at low temperature under $-40$ °C. Surprisingly, the battery can still work well even with the frozen electrolyte. A specific capacity of 134 mAh g$^{-1}$ was obtained at a current density of 0.4 mA cm$^{-2}$ (0.5 C) at $-40$ °C (Fig. 5b). At a higher current density of 4 mA cm$^{-2}$ (5 C), this battery exhibits a specific capacity of 118 mAh g$^{-1}$ (Fig. 5b), still demonstrating a superior rate ability under such low temperature. When decreasing to $-70$ °C, the hydronium-ion battery could still exhibit a specific capacity of

110 mAh g$^{-1}$ at a current density of 0.4 mA cm$^{-2}$ (0.5 C) (Fig. 5c). And a specific capacity of 89 mAh g$^{-1}$ is achieved at a higher current density of 2 mA cm$^{-2}$ (2.5 C), respectively (Fig. 5c). In addition, this battery performs a stable cycle performance at $-70$ °C, remaining a capacity retention above 99% over 100 cycles at a current density of 0.8 mA cm$^{-2}$ (1 C) (Fig. 5d). These facts verify the outstanding low-temperature performance of the versatile PTO//MnO$_2$@GF hydronium-ion battery (see Supplementary Table 3), which is expected to direct the path for exploiting advanced batteries for low-temperature applications, even for outer space use. Moreover, the interesting phenomenon that H$_3$O$^+$ could transfer in a solid-state electrolyte (that is the frozen electrolyte) might shed new light for developing low-temperature solid-state batteries. Certainly, the operation of the PTO//MnO$_2$@GF hydronium-ion battery also depends on the diffusion of Mn$^{2+}$ around cathode. According to the recent report[46], the frozen MnSO$_4$ (1 mol L$^{-1}$) at $-10$ °C exhibits a conductivity of $1.07 \times 10^{-5}$ S cm$^{-1}$, indicating that Mn$^{2+}$ can diffuse in a frozen electrolyte. In our battery, 2 M MnSO$_4$ + 2 M H$_2$SO$_4$ solution is

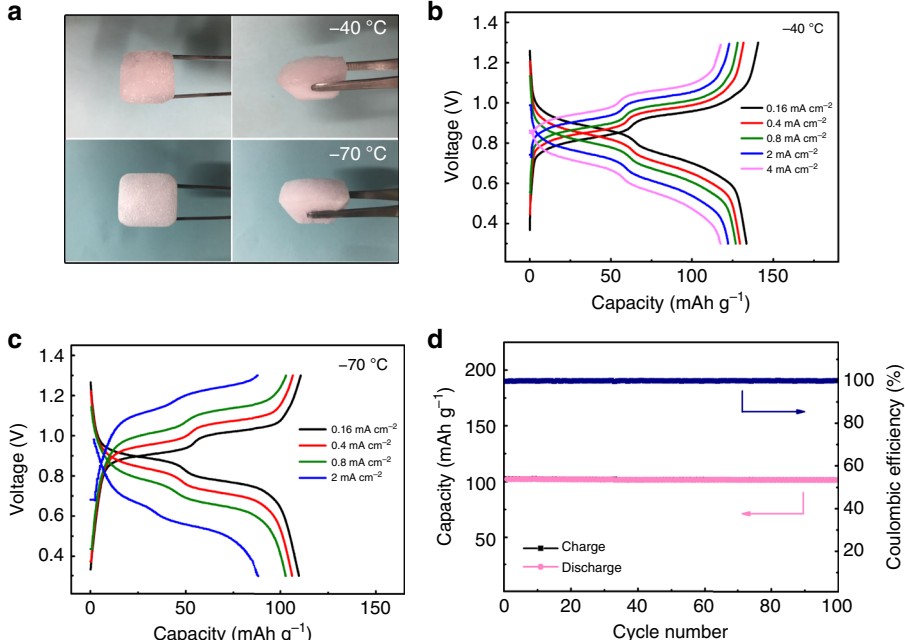

**Fig. 5 Low-temperature behavior investigations of the PTO//MnO₂@GF hydronium-ion battery. a** Optical photographs of the hybrid electrolyte (2 M MnSO₄ + 2 M H₂SO₄) after remaining at −40 °C (above) and −70 °C (below) for several hours. **b, c** Rate ability of the PTO**//**MnO₂@GF hydronium-ion battery described by galvanostatic charge-discharge profiles at various current densities at −40 °C (**b**) and at −70 °C (**c**). **d** Cycle performance at −70 °C of the hydronium-ion battery tested at a current density of 0.8 mA cm⁻² (1 C).

used as electrolyte, where the presence of proton leads to a much lower freezing point of approximately −41.6 °C (Supplementary Fig. 19). As a result, the 2 M MnSO₄ + 2 M H₂SO₄ electrolyte shows a much higher conductivity ($2.66 \times 10^{-3}$ S cm⁻¹) than 2 M MnSO₄ electrolyte ($7.962 \times 10^{-7}$ S cm⁻¹) at −70 °C (see the electrochemical impedance spectroscopy (EIS) results at −70 °C shown in Supplementary Fig. 20 and discussion in Supplementary Note 6). The room-temperature (25 °C) conductivity of the 2 M MnSO₄ + 2 M H₂SO₄ electrolyte is $5.31 \times 10^{-1}$ S cm⁻¹ (see Supplementary Note 7, obtained by EIS data shown in Supplementary Fig. 21), which is only two orders of magnitude higher than the conductivity at −70 °C. However, it is still very difficult to calculate an exact diffusion coefficient of $Mn^{2+}$ in the hybrid electrolyte containing proton, $Mn^{2+}$ and $SO_4^{2-}$, especially at the frozen situation. Ion diffusion in a frozen electrolyte should be a new and complex topic, which needs further investigations in the future.

## Discussion

In this work, we demonstrates a hydronium-ion battery based on an organic PTO-based anode and an inorganic MnO₂@GF cathode, which relies on the quinone/hydroquinone redox reaction on anode and the $Mn^{2+}$/MnO₂ conversion reaction on cathode, along with $H_3O^+$ charge carrier transfer in between in the electrolyte. Owing to the unique electrode reactions, this hydronium-ion battery can integrate high energy and power densities. It exhibits a high energy density of 132.6 Wh kg⁻¹, and a power density up to 30.8 W kg⁻¹. The structure stability and high reversibility of the $H_3O^+$ release/capture process of electrodes contributes to long cycle life over 5000 cycles, along with high Coulombic efficiency approaching 100%. The flexible belt-shaped configuration has also been built with robust mechanical property and superior flexibility. Moreover, the hydronium-ion battery could operate well at low temperature from −40 °C to −70 °C in spite of the frozen electrolyte. The inspiring low-temperature behavior of the versatile hydronium-ion battery

is expected to trigger a new approaching to develop low-temperature and/or solid-state batteries. On the other hand, the organic electrodes generally involve plenty of carbon additives because of their inherent low electronic conductivity, which further reduces the practical energy/power density of corresponding device. This problem should be further investigated in the future.

## Methods

**Synthesis and characterization.** Pyrene-4,5,9,10-tetraone (PTO) was prepared by a previous method[28]. 5 g (25.4 mmol) pyrene (≥99.0%, Sigma-Aldrich) was added into 100 mL CH₂Cl₂ (≥99.9%, Sigma-Aldrich) and 100 mL acetonitrile (≥99.0%, Sigma-Aldrich), then 44.5 g (207.9 mmol) NaIO₄ (≥99.8%, Sigma-Aldrich), 125 mL H₂O and 0.64 g (3.1 mmol) RuCl₃·xH₂O (≥99.98%, Sigma-Aldrich) wan added successively. After heating the mixture at 40 °C overnight, organic solvents were removed through rotary evaporation treatment. Rinsed the solid with H₂O, filtrated, and dried at 70 °C in air. Thereout, dark green solid was obtained. Then, the golden needle-like product (PTO) was obtained by column chromatography treatment (CH₂Cl₂ was used as mobile phase). The morphologies of as-prepared PTO material and PTO-electrodes were characterized by scanning electron microscope (SEM, FEG-S4800). ¹H Nuclear magnetic resonance (NMR) characterization of PTO material was conducted on a 500 MHz NMR spectrometer to define the structure of the as-prepared PTO material. Ex-situ Fourier transform infrared spectroscopy (FTIR) was conducted to examine the structure and purity of as-prepared PTO material and to analyze reversible $H_3O^+$ captured/released mechanism on PTO anode. The elemental distributions of PTO anode and of MnO₂@GF cathode at various charge-discharge states were characterized by scanning transmission electron microscopy (STEM)-Energy dispersive spectrum (EDS) elemental mapping, and SEM-EDS elemental mapping, respectively. X-ray photoelectron spectroscopy (XPS) was carried out to investigate the elemental existence and valent state of PTO anode and MnO₂@GF cathode at various charge-discharge states. X-ray diffraction (XRD) patterns were obtained by Bruker D4 Endeavor X-ray diffractometer, employing Cu-Kα radiation (40 kV, 40 mA), to analyze the structure of electrodeposited MnO₂ on GF cathode. Differential scanning calorimetry (DSC) measurements were conducted to determine the freezing point of electrolytes with a NETZSCH DSC 200 F3 Maia Instrument (Germany). The electrolyte sample was sealed in an aluminum sample pan. The sample was first cooled down to −140 °C at a rate of 5 °C min⁻¹ using liquid nitrogen cooling system, and remained at −140 °C to equilibrate sample temperature, then recovered to 25 °C at a rate of 5 °C min⁻¹. The freezing point was defined as the onset melting point of the endothermal peak of frozen electrolyte.

**Electrode preparation and battery fabrication**. PTO membrane was prepared by mixing active material (i.e., PTO) with conductive additive (Ketjen black, ≥99.9%, Sinopharm) and binder (PTFE, 60 wt.% dispersion in $H_2O$, Sigma-Aldrich) in a mass ratio of 60:30:10 in isopropanol (≥99.7%, Sigma-Aldrich) solvent. Activated carbon (AC) membrane was prepared by mixing activated carbon (≥99.999%, Sigma-Aldrich) with Ketjen black and PTFE in a mass ratio of 85:5:10 in isopropanol. The active material membrane was obtained by rolling the slurry into a membrane and going through drying treatment at 120 °C overnight in air. For electrochemical tests in typical three-electrode systems, PTO and AC electrodes were produced by suppressing required active material membrane on titanium mesh current collector. The active material mass loading of PTO electrode was about $4\,mg\,cm^{-2}$ with an electrode area of $0.6\,cm^2$. A piece of GF ($1.0\,cm^2$) with pre-electrodeposited 5 mAh $MnO_2$ ($MnO_2@GF$) was used as cathode connected with a platinum electrode holder for the individual electrochemical performance test. It should be noted that GF was pre-treated to charge at a constant current density of $5\,mA\,cm^{-2}$ for 1 hr using a three-electrode system to enhance the coulombic efficiency of the full battery. Activated carbon and Ag/AgCl electrodes were used as counter and reference electrodes for three-electrode systems, respectively. 2 M $MnSO_4$ + 2 M $H_2SO_4$ hybrid aqueous solution was used as electrolyte. For assembling the hydronium-ion full battery, the as-prepared PTO electrode and $MnO_2@GF$ electrode were employed as anode and cathode, respectively. To demonstrate the flexible belt-shaped configuration, PTO membrane and GF were cut in size of $1\,cm \times 5\,cm$, respectively. Then electrodes were prepared by connecting PTO membrane pressed on soft titanium mesh with nickel lugs, respectively. The active material mass loading of flexible PTO electrode was about $6\,mg\,cm^{-2}$. Glass fiber (Waterman Glass microfiber filter) was used as separator. PTO anode, glass fiber separator and $MnO_2@GF$ cathode were stacked up to form a "sandwich" structure. The belt-shaped aqueous PTO//$MnO_2@GF$ hydronium-ion battery was obtained through a vacuum sealed process after adding 2 M $MnSO_4$ + 2 M $H_2SO_4$ aqueous solution as electrolyte.

**Electrochemical tests**. The electrochemical tests of PTO anode and $MnO_2@GF$ cathode were carried out with a typical three-electrode system using AC electrode and Ag/AgCl electrode as counter and reference electrodes, respectively. 2 M $MnSO_4$ + 2 M $H_2SO_4$, 2 M $MnSO_4$, and 2 M $H_2SO_4$ solutions were used as electrolyte, respectively. Cyclic voltammetry (CV) and galvanostatic discharge/charge methods were conducted to investigate the electrochemical performances of PTO anode, $MnO_2@GF$ cathode, and the full battery. CV curves of PTO-electrode and PTO//$MnO_2@GF$ full battery were obtained in the potential between 0 V and 0.7 V vs. Ag/AgCl and between 0 V and 1.3 V vs. Ag/AgCl, respectively, at a scan rate from 0.1 to $0.5\,mV\,s^{-1}$ on CH Instruments electrochemical workstation (CHI 660D). The galvanostatic discharge/charge measurements were performed on Hukuto Denko battery charge/discharge system HJ series (Japan) controlled computer. The mechanism demonstration tests were carried out in a three-electrode system, with $MnO_2@GF$ electrode and Ag/AgCl electrode as counter and reference electrodes, respectively. A Meiling Biology & Medical DW-HW50 ultra-low freezer was used for low temperature performance investigations of the hydronium-ion battery. The pictures of electrolytes at −40 °C and −70 °C were taken quickly in air at room temperature (25 °C) after being frozen in corresponding temperatures for several hours. The batteries were kept at a specified temperature for 2 h before charge-discharge tests to ensure the battery temperature equivalent to the setting freezer temperature. The batteries were charged-discharged at various current densities at −40 °C and −70 °C, respectively. It should be noted that when tested at −70 °C, the hydronium-ion battery was firstly charged-discharged for several cycles at −65 °C, then tested at −70 °C with the same operation process at −40 °C. A short movie was given for clearly displaying the electrochemical test of the battery at −70 °C (see Supplementary Movie 1). The ion conductivity of electrolyte was investigated by electrochemical impedance spectroscopy (EIS) technology using a CHI instrument combined with two parallel Pt electrodes ($1\,cm \times 1\,cm$) as electrodes in a polytetrafluorethylene (PTFE) cylindrical cell. The measuring frequency range was kept between 0.1 Hz and $10^5$ Hz.

## Data availability

The authors declare that all the relevant data are available within the paper and its Supplementary Information file or from the corresponding author upon reasonable request.

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

## Acknowledgements

The authors acknowledge funding support from the National Natural Science Foundation of China (21975052, 21622303) and the State Key Basic Research Program of China (2018YFE0201702, 2016YFA0203302).

## Author contributions

Y.W. conceived and designed the experiments. Z.G. prepared the materials, fabricated the battery, and carried out experiments. J.H., L.Y. and Z.W. assisted Z.G. in materials characterization. X.D. and Y.X. gave some suggestions about the experiments. Z.G. and Y.W. co-wrote the paper. All authors discussed the results and commented on the manuscript.

## Competing interests

The authors declare no competing interests.
