## [Peer Review File · Nature Communications]

Reviewers' comments:

Reviewer #1 (Remarks to the Author):

This work reports the design of a proton-based aqueous rechargeable battery using inorganic MnO₂ as the cathode and organic PTO as the anode. While the reported performance is improved compared to previous reports in this area, I have some major concerns regarding the reaction mechanisms which are currently not clear. Therefore, I cannot recommend publication in NC. Detailed comments are as following:

1. The authors discuss lots of merits of H⁺-based battery, one of which is the smallest radius of H⁺ compared to other cations such as Li⁺ and Na⁺ and H₃O⁺. However, it seems that the actively inserted ions here in this work are still H₃O⁺, not H⁺. So this leads the reviewer, and possibly the future readers to wonder how the mentioned drawbacks of H₃O⁺ in the introduction are addressed in this work. Apparently to the reviewer, this is not addressed, which results in my doubt about the novelty of this work. At least, the authors should avoid discussing the H⁺-based battery, which is quite misleading. Using the term H₃O⁺ instead of H⁺ (or proton) in both the title and the main text (including Scheme 1) will be more appropriate.

2. Regarding the Equation 1, it seems that the authors provide enough evidence to confirm the anode side reaction mechanisms. However, there is no evidence showing that the cathode MnO₂ phase can be reversely taking part in the reaction. For example, there is no study confirming MnO₂ can disappear from the cathode during the ion insertion reaction. More importantly, MnO₂ has been known to possess large opening framework for ion intercalation, and it experiences framework-drive ion intercalation reaction followed by conversion reaction with a phase transition, and such phase transition is typically not reversible. Saying that, the authors need to provide extra evidences to clarify the MnO₂ cathode reaction mechanisms as described in the discussion.

3. In Figure 2, please show more STEM-EDS results for gradual MnO₂ loading experiments under different current. Can the authors please also provide the mass loading for MnO₂ on the current collector? The reviewer believes this is very important to relate this to the overall capacity of the battery during cycling.

4. The low-temperature performance is really interesting. Can the authors compare their value to other reported state-of-the-art values of low-T battery studies to demonstrate how good is good?

5. Provide the pH values of the electrolyte, which is claimed by the authors to be a constant value. And also please explain how the value being constant during cycling is confirmed.

Reviewer #2 (Remarks to the Author):

In this paper the authors present a secondary battery composed of a quinone-type anode (pyrene-4,5,9,10-tetraone) (PTO) and a graphite felt cathode (pre-deposited with MnO₂) serving as substrate for oxidative deposition of Mn²⁺ forming MnO₂. A 2M H₂SO₄ water solution containing 2M MnSO₄ is used as electrolyte and the authors claim that this battery operates according to a rocking-chair principle. Both the PTO anode and the carbon felt cathode are characterized in 3-electrode set-up and both electrodes show excellent rate capability and cycling stability. These promising properties are also retained when the electrodes are combined into complete cells. In addition, the battery seems to perform well also at very low temperatures (-40 centigrade and -70 centigrade) as well as under mechanical stress. The presented results are thus spectacular and will likely have a profound impact on future research in the field of organic-, water-based- and flexible batteries. However there are several points that need to be clarified before publication and this reviewer therefor recommend publication after major revision. Specific points are listed below with major issues presented first and minor issues later:

Main concerns:

1) The redox reaction occurring at the PTO anode is depicted in scheme 1. From the figure it is

clear that the authors claim the complete reaction to be a 4e4H process. There is, however, insufficient proof of this over-all reaction in the manuscript. Some data rather indicate that the complete reaction is a 2e2H reaction: Based on the assumed 4e process and the molecular mass of PTO the theoretical capacity should be around 400 mAh/g. In both 3-electrode set-up and in 2 electrode set-up the evaluated capacity is slightly more than half the theoretical value. In addition, the IR spectroscopy performed at different state of charge shows that even in the fully reduced form there is absorption corresponding to carbonyl vibration remaining. The claimed redox reactions occurring at the PTO anode thus need further support.

2) In figure 3C the authors use a Ragone plot to benchmark the presented battery against existing technologies. Although this is often used in the literature it is instrumental for such comparison that the figures of merit are compared using the same normalization procedure. It is likely that the data from existing technologies used for comparison are on device level and hence that the normalization is against the device mass. The capacity and power of the material presented in this report, on the other hand, is normalized against the weight of active material alone. (Using the energy density provided, 170.5 Wh/kg (line 160), the specific capacity can be estimated to 213 mAh/g ($C/m = E_{sp} \cdot 1000/U$) using an average output voltage (U) of 0.8 V. This is close to the capacity of the PTO electrode and hence it is clear that the authors use only active material weights for normalization). In doing so the mass of carbon additives and binders that make up for 40% of the weight of the PTO membrane is ignored and so is 1) the mass of the electrolyte (containing the Mn²⁺ reservoir), 2) the titanium mesh used as current collector, 3) the MnO₂ coated carbon mesh, 4) the separator and 5) packaging. This makes the comparison with existing technologies misleading.

3) The IR-spectra presented in Supplementary figure 8 looks highly distorted with unsymmetrical peaks suggesting that these spectra have been manipulated. Such manipulation must be described somewhere in the manuscript or provided as supporting information. In addition the interpretation of the IR-data could be improved in support of the claimed redox processes occurring. It would, for instance, be helpful to see how key transitions/absorptions evolve as function of potential.

4) The most spectacular feature of the presented battery is the low temperature behavior. As briefly discussed in the presented manuscript, proton diffusion in frozen water is known to be relatively fast and, hence, as indicated in line 187-194 a proton rocking chair battery may function also in frozen media. However, the presented battery is not a proton rocking chair battery (see below). Instead the redox reaction at the cathode side require diffusion of both water molecules (and hydronium ions) and Mn²⁺ ions. The feasibility of such mass transport should be investigated in addition to the low-temperature battery tests or precise references be added. I.e. what is the diffusion coefficient of Mn²⁺ in the frozen electrolyte used?

Minor comments:

Rocking-chair type batteries are characterized by the condition that the electrolyte does not have to accommodate excess ions necessary for the redox reactions occurring at the electrodes. In the present study the electrolyte serve as reservoir for electroactive Mn²⁺ ions and this reviewer therefor find it misleading to place the present battery under the rocking-chair category of batteries.

Line 83-84: A linear increase of peak current with scan rate does not necessarily indicate fast transport. Also, in the corresponding supplementary figure 5 the use of "capacitive behavior" is misleading as the redox processes investigated are certainly not capacitive in nature nor is the current independent of potential as indicative of a capacitive response. Also the discussion in relation to figure "Supplementary figure 9" is confusing. First of all, a slope of 0.61 is clearly far from what is expected for a Nernstian surface confined reaction (as is one proper way to describe a reaction with a linear $i_p = f(\text{scan rate})$). Second, a surface confined reaction cannot be expected as the reaction relies on diffusion of Mn²⁺ towards the electrode surface. Third, as the geometry is much more complicated than the authors imply a diffusion limited reaction cannot be expected to follow a simple square-root-dependence.

Line 99, 100, and 110: The use of charge/discharge to describe oxidation/reduction of PTO is

confusing as the opposite reaction corresponds to charge and discharge in the battery cell.

Line 125: The statement that the coulombic efficiency is close to 100 % is clearly incorrect (as seen in figure 2b)

Line 192-193: It is unclear what the authors mean by "Herein, the PTO//MnO₂@GF proton cell shows a distinct characteristic in absence of desolvation process of H₃O⁺, thus might be able to work well at low temperature".

Line 260: In addition to the areal mass loadings, please provide the electrode area used in the experiments.

A general improvement of the language is recommended.

Reviewer #3 (Remarks to the Author):

In present work, Guo et al reported a rocking-chair-type proton battery based on an organic PTO anode and MnO₂@graphite felt anode, which works upon quinone/hydroquinone redox reaction on anode and MnO₂/Mn²⁺ conversion reaction on cathode. In consideration of electrode materials exploration for proton batteries, the intercalation-type inorganic electrode materials exhibit limited performance due to the large ion radii of hydronium ions, the existing form of proton in aqueous solution. In this work, it is a very ingenious breakthrough to utilize organic/inorganic battery system based on enolization and conversion reactions rather than intercalation mechanism, thus avoiding the inherent slow kinetics and inspiring the wide choices of hydronium ion host. Such proton batteries deliver high energy density and high power density comparable to supercapacitors, as well as good cycle stability. The fabricated flexible configuration also shows good performance with outstanding energy density compared to the state-of-art reports. While the most interesting and exciting result is that the developed proton battery could work normally under such low-temperature as minus 70 degrees, which performs as a powerful member for low-temperature batteries, and might bring more inspiring research on development of low-temperature power sources. Overall, this work is interesting and important, which can be recommended for the publication in Nature Communications after minor revision. The minor concerns are summarized as following:

1. The theoretical specific capacity of PTO material should be calculated as about 400 mAh/g based on four carbonyl groups per PTO molecular. While the delivered specific capacity of PTO in this work is only 208 mAh/g. Why does PTO exhibit only a half of the theoretical capacity here?
2. Both traditional and flexible configurations of the PTO-based proton batteries have been fabricated in this work. The mass loading of active material on cathode (i.e., PTO) for these two configurations should be given for better understanding.
3. Although such battery exhibits some inherent advantages, its disadvantages, i.e., low electronic conductivity of organic electrode, i.e., PTO, should also be mentioned in this paper.
4. In Figure 2, the authors show a lot of ex-situ characterization results of MnO₂@GF cathode, e.g. ex-situ XPS spectra and XRD patterns. However, the test conditions of corresponding electrochemical measurements (such as applied currents) are not given. Please clarify this point in main-text or experimental section.
5. When discussing the energy and power densities, the mass which calculation based on should be clearly defined, e.g., on both electrodes or not? This should be made clear when comparing to other reports, like in supplementary Table 1.
6. The optical pictures of electrolyte under -40 and -70 degrees were given in Figure 4a. The real time temperature when the pictures were taken should be given clearly for precisely reflecting the actual state of electrolyte.

Response to Reviewer#1

Overall Comment: This work reports the design of a proton-based aqueous rechargeable battery using inorganic MnO_2 as the cathode and organic PTO as the anode. While the reported performance is improved compared to previous reports in this area, I have some major concerns regarding the reaction mechanisms which are currently not clear. Therefore, I cannot recommend publication in *NC*. Detailed comments are as following:

Response: Thanks for reviewing our manuscript. We would like to answer your questions separately and revise the manuscript according to your suggestions. All the revisions according to your questions/suggestions have been highlighted by cyanine background in the revised manuscript.

Question-1: The authors discuss lots of merits of H^+ -based battery, one of which is the smallest radius of H^+ compared to other cations such as Li^+ and Na^+ and H_3O^+ . However, it seems that the actively inserted ions here in this work are still H_3O^+ , not H^+ . So this leads the reviewer, and possibly the future readers to wonder how the mentioned drawbacks of H_3O^+ in the introduction are addressed in this work. Apparently to the reviewer, this is not addressed, which results in my doubt about the novelty of this work. At least, the authors should avoid discussing the H^+ -based battery, which is quite misleading. Using the term H_3O^+ instead of H^+ (or proton) in both the title and the main text (including Scheme 1) will be more appropriate.

Response: According to your suggestion, we would like to use the term H_3O^+ instead of H^+ (or proton) in both the title and the main-text (including Scheme 1). In fact, the purpose of this work is to highlight the merits of H_3O^+ battery (PS: please see equation-1, 2 in our original submission), rather than its drawbacks. You might misunderstand our description in the second paragraph of the main-text. What we want to present is further explained as follows:

(1) The naked H^+ can rarely be used as charge carrier because of the formation of hydronium ions (H_3O^+), where the high dehydration energy of H_3O^+ (11.66 eV) prevent the de-solvent process ($\text{H}_3\text{O}^+ \rightarrow \text{H}^+ + \text{H}_2\text{O}$). That is to say, H_3O^+ can generally be used as the charge carrier, rather than the naked H^+ .

(2) It is undoubted that the hydronium (H_3O^+) charge carrier shows the cost effectivity and sustainability advantages over these metal ions. However, only several hydronium-ion batteries were demonstrated recently, which might be attributable to that the size of H_3O^+ (that is larger than naked Li^+ and close to naked Na^+) limit choice of host materials (that are the intercalation compounds). Here, we demonstrate a new hydronium-ion battery with improved performance (e.g. higher H_3O^+ storage capacity and longer life), which shows the advantages of hydronium-ion battery to readers.

(3) Especially, the storage of H_3O^+ in electrode without de-solvent process (i.e. the process of H_3O^+

→ $H^+ + H_2O$ at interface) might enable perfect low temperature performance. It is well known that the de-solvent process at interface limits the kinetics and especially the low temperature performance of electrodes. However, such potential merit of hydronium-ion battery has never been reported. Here, for the first time, we demonstrate the perfect low temperature performance for popular readers.

In summary, our purpose is to introduce the merits of hydronium-ion battery. **According to your suggestion, we have used the term H_3O^+ instead of H^+ (or proton) in both the title and the main-text (including Scheme 1). Furthermore, we also deleted the discussion about H^+ -based battery and improve our description in the second paragraph and the discussion section of the main-text.**

Question-2: Regarding the Equation 1, it seems that the authors provide enough evidence to confirm the anode side reaction mechanisms. However, there is no evidence showing that the cathode MnO_2 phase can be reversely taking part in the reaction. For example, there is no study confirming MnO_2 can disappear from the cathode during the ion insertion reaction. More importantly, MnO_2 has been known to possess large opening framework for ion intercalation, and it experiences framework-drive ion intercalation reaction followed by conversion reaction with a phase transition, and such phase transition is typically not reversible. Saying that, the authors need to provide extra evidences to clarify the MnO_2 cathode reaction mechanisms as described in the discussion.

Response: Thanks for your question. Herein, we would like to answer you as follows:

(1) According to your suggestion, SEM images of the electrode (pristine, after charge and after discharge) have been compared to demonstrate the dissolution of MnO_2 on discharge (i.e. MnO_2 to Mn^{2+}). Please see **Figure answer 1** or **Supplementary Figure 15** in revised supplementary information.

Figure answer 1. SEM images for GF electrodes at pristine, charged (1 mAh, 5 mAh, 10 mAh MnO₂ deposition capacities), and discharged states with a current density of 5 mA cm⁻².

(2) It is well known that the discharge of MnO₂ in acid includes the ion intercalation (electrochemical process) and the disproportionated reaction of Mn³⁺ (chemical process), through which MnO₂ is converted into Mn²⁺ (e.g. *J. Electrochem. Soc.*, 2012, 159, A1554-A1561). Generally, the discharge can be summarized as:

The overall reaction of Pathway-1 and Pathway-2 is the same, which can be given as:

In fact, the reversible deposition/dissolution of MnO₂ in acid has been well clarified by many previous reports (e.g. *J. Electrochem. Soc.*, 2012, 159, A1554-A1561, *J. Appl. Electrochem.*, 1998, 28, 1235-1241, *J. Electroanal. Chem.*, 2011, 651, 237-242, *Electrochimica Acta*, 2007, 52, 4630-4639, etc.) According to your question, we have improved the discussions about the reaction mechanism of MnO₂ deposition/dissolution in cathode with proper citations. In addition, the above equations (1-5) are also given as extended discussion in the **Supplementary Figure 15**. Herein, it also should be noted that our work focuses on “combining the reversible deposition/dissolution of MnO₂ in cathode and C=O/C-O- conversion in anode by using H₃O⁺ charge carrier, in order to

demonstrate the advantages of hydronium-ion battery (especially the low temperature performance)”.

Question-3: In Figure 2, please show more STEM-EDS results for gradual MnO₂ loading experiments under different current. Can the authors please also provide the mass loading for MnO₂ on the current collector? The reviewer believes this is very important to relate this to the overall capacity of the battery during cycling.

Response: Thanks for your question. Herein, we would like to answer your questions as follows:

(1) The electrochemical deposition of MnO₂ in acid has been widely used for large-scale MnO₂ production, and related mechanism was well clarified (please also see our response to Question-2). According to your kind suggestion, we further investigated the morphologies of the electrode after MnO₂ deposition with different currents and capacities by SEM-EDS analysis. As shown in **Figure answer 2**, the MnO₂ deposition layer is coated on the outer surface of carbon electrode to form a coaxial cable structure. (By the way, in our original submission, the caption of **Figure 2d** should be SEM and SEM EDS-elemental mapping images.) **Figure answer 2** has been given in the revised supplementary information as **Supplementary Figure 13**.

Figure answer 2. SEM-EDS elemental mapping images of GF-based electrodes with various MnO₂ deposition capacities of 1 mAh cm⁻² (with a current of 2 mA cm⁻²), 5 mAh cm⁻² (with a current of 10 mA cm⁻²), and 25 mAh cm⁻² (with a current of 50 mA cm⁻²).

(2) The loading of MnO₂ depends on the charge depth. According to $It = nZF$, 1 mAh cm⁻² charge capacity indicates a typical mass loading of 1.6 mg cm⁻². It is undoubted that the cathode and anode in a battery should keep capacity balance. Therefore, we control the charge depth (or mass loading) of MnO₂ by the mass of organic anode (that is PTO). In our submission, the mass loading of PTO in

anode is 4 mg cm^{-2} , which shows a capacity of $0.832 \text{ mAh cm}^{-2}$ ($= 4 \times 10^{-3} \text{ g cm}^{-2} \times 208 \text{ mAh g}^{-1}$). According to this value ($0.832 \text{ mAh cm}^{-2}$), the loading of MnO_2 in cathode over a charge/discharge cycle is 1.33 mg cm^{-2} . Certainly, we can increase the loading MnO_2 in cathode by enhancing the mass loading PTO in anode. To clarify this point, we also fabricate a battery using a much higher mass loading (25 mg cm^{-2}) of PTO in anode (see **Supplementary Figure 16** in the revised supplementary information). With such PTO anode, the mass loading of MnO_2 can reach 8 mg cm^{-2} (**Supplementary Figure 16**). Certainly, to some extent, the high mass loading slightly limits the utilization of PTO (see the discussion about **Supplementary Figure 16** in the revised supplementary information).

Question-4: The low-temperature performance is really interesting. Can the authors compare their value to other reported state-of-the-art values of low-T battery studies to demonstrate how good is good?

Response: According to your suggestion, the achieved low temperature performance is compared with the recent achievements about low temperature batteries, including *Science* 2017, 356, 6345 ($-60 \text{ }^\circ\text{C}$ with liquefied gas electrolyte)/ *Nature* 2016, 529, 515 ($-40 \text{ }^\circ\text{C}$ using organic electrolyte)/ *Joule* 2018, 2, 902 (Our recent work, $-70 \text{ }^\circ\text{C}$ using organic electrolyte)/ *Angew. Chem. Int. Ed.* 2019, 58, 16994 ($-50 \text{ }^\circ\text{C}$ using organic + aqueous electrolyte). Please see **Supplementary Table 3** in the revised supplementary information for detailed information. It is undoubted that our result among the best achievements. Especially, aqueous electrolyte operated at $-70 \text{ }^\circ\text{C}$ has been never reported.

Question-5: Provide the pH values of the electrolyte, which is claimed by the authors to be a constant value. And also please explain how the value being constant during cycling is confirmed.

Response: Thanks for your good question. We are sorry for mistakenly describing the charge/discharge process of this battery as a pH-constant process. Operation mechanism of the battery is summarized as follows:

On charge, the amount of generated H_3O^+ (four) in cathode is more than that of stored H_3O^+ (two) in anode, resulting the H_3O^+ amount increase in the electrolyte during charge. Discharge reverses charge process with H_3O^+ amount decrease to the initial value. That is to say, the concentration of H_3O^+ changes over charge/discharge process. According to your question, the corresponding

description about pH-constant process has been deleted. In addition, according to the comments of reviewer # 2, the related description about rocking chair operation of the battery has also be deleted (Please see our response to Reviewer # 2 for details).

—

Response to Reviewer #2

Overall Comment: In this paper the authors present a secondary battery composed of a quinone-type anode (pyrene-4,5,9,10-tetraone) (PTO) and a graphite felt cathode (pre-deposited with MnO_2) serving as substrate for oxidative deposition of Mn^{2+} forming MnO_2 . A 2M H_2SO_4 water solution containing 2M MnSO_4 is used as electrolyte and the authors claim that this battery operates according to a rocking-chair principle. Both the PTO anode and the carbon felt cathode are characterized in 3-electrode set-up and both electrodes show excellent rate capability and cycling stability. These promising properties are also retained when the electrodes are combined into complete cells. In addition, the battery seems to perform well also at very low temperatures (-40 centigrade and -70 centigrade) as well as under mechanical stress. The presented results are thus spectacular and will likely have a profound impact on future research in the field of organic-, water-based- and flexible batteries. However there are several points that need to be clarified before publication and this reviewer therefor recommend publication after major revision. Specific points are listed below with major issues presented first and minor issues later:

Response: Many thanks for kindly reviewing our manuscript and giving us plenty of very useful suggestions. In corresponding to each question pointed out by you, we would like to answer separately and carefully revise the manuscript according to your suggestions. All the revisions according to your questions/suggestions have been highlighted by yellow background in the revised manuscript.

Main concerns:

Question-1: The redox reaction occurring at the PTO anode is depicted in scheme 1. From the figure it is clear that the authors claim the complete reaction to be a 4e4H process. There is, however, insufficient proof of this over-all reaction in the manuscript. Some data rather indicate that the complete reaction is a 2e2H reaction: Based on the assumed 4e process and the molecular mass of PTO the theoretical capacity should be around 400 mAh/g. In both 3-electrode set-up and in 2 electrode set-up the evaluated capacity is slightly more than half the theoretical value. In addition, the IR spectroscopy performed at different state of charge shows that even in the fully reduced form there is absorption corresponding to carbonyl vibration remaining. The claimed redox reactions

occurring at the PTO anode thus need further support.

Response: Thanks for your good question. As correctly pointed out by you, the discharge capacity of PTO electrode in three-electrode test or full battery test at a low current density is only approximately 200 mAh g⁻¹, which is about half of the theoretical capacity (approximately 400 mAh g⁻¹) calculated based on 4e4H reaction. That is to say, only ~ 50% carbonyl groups can be used for charge storage, indicating a 2e2H reaction. As mentioned by you, the IR spectroscopy performed at different states of charge shows that even in the fully reduced form there is absorption corresponding to carbonyl vibration remaining, which confirms the above conclusion. The limited utilization might be attributable to that the initial H₃O⁺ combination on the carbonyl groups leads to the large steric hindrance for the further H₃O⁺ combination on adjacent carbonyl groups. According to your question, we have changed scheme 1 as a 2e2H reaction, and revised the related equation/discussion as above, see these sentences highlighted by yellow background in page 2, 3, 4 and 5 of revised manuscript.

Question-2: In figure 3C the authors use a Ragone plot to benchmark the presented battery against existing technologies. Although this is often used in the literature it is instrumental for such comparison that the figures of merit are compared using the same normalization procedure. It is likely that the data from existing technologies used for comparison are on device level and hence that the normalization is against the device mass. The capacity and power of the material presented in this report, on the other hand, is normalized against the weight of active material alone. (Using the energy density provided, 170.5 Wh/kg (line 160), the specific capacity can be estimated to 213 mAh/g ($C/m = E_{sp} \cdot 1000/U$) using an average output voltage (U) of 0.8 V. This is close to the capacity of the PTO electrode and hence it is clear that the authors use only active material weights for normalization). In doing so the mass of carbon additives and binders that make up for 40% of the weight of the PTO membrane is ignored and so is 1) the mass of the electrolyte (containing the Mn²⁺ reservoir), 2) the titanium mesh used as current collector, 3) the MnO₂ coated carbon mesh, 4) the separator and 5) packaging. This makes the comparison with existing technologies misleading.

Response: Many thanks for this very good comment. We quite agree with you that the comparison in **Figure 3c** is misleading. (1) We are sorry for such comparison, and thus we deleted **Figure 3c** in the revised manuscript to eliminate this mistake. (2) In the revised supplementary information, a

table (i.e. **Supplementary Table 1**) is used to compare the achieved energy/power density with refs. 31-35 (about H_3O^+ battery) that are calculated based on the electrode active materials. (3) In the revised manuscript, we re-calculated the energy/power density based on both the mass of PTO and the mass of deposited (or dissolved) MnO_2 during charge/discharge cycles (see **Supplementary Table 1** or the related discussion in the main-text). We also emphasize that the energy/power density is only calculated based on active materials for comparison. When converted into practical devices, a great discount should be considered for these calculated values because of the additional materials of a practical device, including current collector, separator, electrolyte, package and so on (see the discussion of **Supplementary Table 1**). (4) In the discussion section, we also mentioned that the organic electrodes generally involve plenty of carbon additives because of their low electronic conductivity, which further reduces the practical energy/power density of corresponding device. This problem should be further investigated in the future.

Question-3: The IR-spectra presented in Supplementary figure 8 looks highly distorted with unsymmetrical peaks suggesting that these spectra have been manipulated. Such manipulation must be described somewhere in the manuscript or provided as supporting information. In addition, the interpretation of the IR-data could be improved in support of the claimed redox processes occurring. It would, for instance, be helpful to see how key transitions/absorptions evolve as function of potential.

Response: Many thanks for your very good suggestions. (1) As corrected pointed out by you, the IR-spectra (Figure 1e and Supplementary Figure 8) were manipulated by a routine baseline correction and background removal procession for the clear observation of characteristic absorptions, which have been explained in the main-text of revised manuscript and in the caption of **Supplementary Figure 8**, marked by yellow highlight. (2) According to your kind suggestion, the detailed interpretation of IR-spectra related to the redox process has been added into the main-text with yellow highlights (see page 4 in revised manuscript) for distinct understanding of the relationship between characteristic absorptions of carbonyl groups and potential evolvement of PTO-based electrode. [See page 4: The characteristic absorption peaks at $\sim 1670\text{ cm}^{-1}$ is ascribed to carbonyl groups of PTO molecule, whose intensity is positively correlated with the content of pristine carbonyl groups. During the reduction (discharge) process, the reduction depth of active groups on PTO increases along with the potential decrease. While during the oxidation (re-charge) process, the oxidation depth of active groups increases with the potential increase. Therefore, the content of redox-engaged carbonyl groups reflected by absorption peak intensity will evolve with discharge/charge potential. As is shown in

Figure 1e, the peak intensity of at $\sim 1670\text{ cm}^{-1}$ decreases with the reduction depth increase of PTO-electrode. Subsequently, these peaks strengthen back gradually to approach that of pristine state during re-charge process (**Figure 1e**), confirming the reversible quinone/hydroquinone redox reaction occurring on PTO-electrode (see **Supplementary Figure 8** for the full FT-IR spectra). It should be noted that the IR-spectra shown in **Figure 1e** and **Supplementary Figure 8** were manipulated by a routine baseline correction and background removal process for the clear observation of characteristic absorptions. In addition, the FT-IR spectra performed at different state of oxidation show that even in the fully reduced form there is absorption corresponding to carbonyl vibration remaining, indicating that not all of carbonyl groups can be used for charge storage (that consists with charge/discharge results).]

Question-4: The most spectacular feature of the presented battery is the low temperature behavior. As briefly discussed in the presented manuscript, proton diffusion in frozen water is known to be relatively fast and, hence, as indicated in line 187-194 a proton rocking chair battery may function also in frozen media. However, the presented battery is not a proton rocking chair battery (see below). Instead the redox reaction at the cathode side require diffusion of both water molecules (and hydronium ions) and Mn^{2+} ions. The feasibility of such mass transport should be investigated in addition to the low-temperature battery tests or precise references be added. I.e. what is the diffusion coefficient of Mn^{2+} in the frozen electrolyte used?

Response: Thank for your good suggestion. Here we would like to answer as follows:

(1) We agree with you that this battery is not a real proton rocking chair battery, and thus we deleted the related description/discussion about rocking chair battery in the revised manuscript.

(2) As correctly pointed out by you, the cathode reaction (i.e. MnO_2 deposition/dissolution) requires the diffusion of both Mn^{2+} ions and water molecules/hydronium ions. According to the recent report (Angew. Chem. Int. Ed. 2019, 58, 12569-12573), the frozen MnSO_4 (1 mol L^{-1} , at $-10\text{ }^\circ\text{C}$) exhibits a conductivity of $1.07 \times 10^{-5}\text{ S cm}^{-1}$. Based on this value, we estimate the diffusion coefficient through Nernst-Einstein equation (*Equation 1*).

$$\sigma = \frac{N_A}{k_B T} \sum_i^n c_i q_i^2 D_i \quad (1)$$

Where N_A is the Avogadro number, e the elementary charge, K_B the Boltzmann constant, T the absolute temperature of the system, and c_i , q_i , D_i , respectively, the concentration, charge, and

diffusion coefficient of ions of species i (cations or anions). For convenience, we consider $D_+ \approx D_-$ (just for estimation). The calculated diffusion coefficient of Mn^{2+} in a frozen electrolyte is about $3.152 \times 10^{-13} \text{ cm}^2 \text{ S}^{-1}$.

(3) Besides citing previous report, we also employed the Electrochemical Impedance Spectroscopy (EIS) measurements to clarify the conductivity of the frozen ($\text{H}_2\text{SO}_4 + \text{MnSO}_4$) electrolyte and MnSO_4 electrolyte at $-70 \text{ }^\circ\text{C}$. As is shown in **Figure answer 3**, the interface resistance of the frozen $\text{H}_2\text{SO}_4 + \text{MnSO}_4$ solution is much lower than that of the frozen MnSO_4 solution (without H_2SO_4), where the lower resistance is attributable to the fast diffusion of H_3O^+ . According to the EIS data, the frozen

Figure answer 3. Nyquist plot of (a) $2 \text{ mol L}^{-1} \text{ MnSO}_4 + 2 \text{ mol L}^{-1} \text{ H}_2\text{SO}_4$ and (b) $2 \text{ mol L}^{-1} \text{ MnSO}_4$ solutions at $-70 \text{ }^\circ\text{C}$.

($\text{H}_2\text{SO}_4 + \text{MnSO}_4$) electrolyte and MnSO_4 electrolyte show the conductivities of $2.66 \times 10^{-3} \text{ S cm}^{-1}$ and $7.962 \times 10^{-7} \text{ S cm}^{-1}$, respectively. For the frozen MnSO_4 electrolyte, the diffusion coefficient of Mn^{2+} in a frozen electrolyte ($-70 \text{ }^\circ\text{C}$) is calculated as $9.052 \times 10^{-15} \text{ cm}^2 \text{ S}^{-1}$ according to the Nernst-Einstein equation (*Equation 1*). Herein, it should be noted that the calculated diffusion coefficient of Mn^{2+} in a frozen MnSO_4 electrolyte at $-70 \text{ }^\circ\text{C}$ cannot represent the real diffusion coefficient of Mn^{2+} in the frozen ($\text{H}_2\text{SO}_4 + \text{MnSO}_4$) electrolyte at $-70 \text{ }^\circ\text{C}$, because the existence of large amount of H_3O^+ might change the state of the electrolyte solution (such as freezing point).

Minor comments:

Rocking-chair type batteries are characterized by the condition that the electrolyte does not have to accommodate excess ions necessary for the redox reactions occurring at the electrodes. In the present study the electrolyte serves as reservoir for electroactive Mn^{2+} ions and this reviewer therefor find it

misleading to place the present battery under the rocking-chair category of batteries.

Response: Thank you for your good comments. We agree with your opinions and the “rocking chair” appellation is abandoned in the revised manuscript.

Question-5: Line 83-84: A linear increase of peak current with scan rate does not necessarily indicate fast transport. Also, in the corresponding supplementary figure 5 the use of “capacitive behavior” is misleading as the redox processes investigated are certainly not capacitive in nature nor is the current independent of potential as indicative of a capacitive response. Also the discussion in relation to figure “Supplementary figure 9” is confusing. First of all, a slope of 0.61 is clearly far from what is expected for a Nernstian surface confined reaction (as is one proper way to describe a reaction with a linear $i_p = f(\text{scan rate})$). Second, a surface confined reaction cannot be expected as the reaction relies on diffusion of Mn^{+2} towards the electrode surface. Third, as the geometry is much more complicated than the authors imply a diffusion limited reaction cannot be expected to follow a simple square-root-dependence.

Response: Many thanks for your very good suggestions. According to your kind suggestions, we revised the manuscript as: (1) the related discussion (Line 83-84 in original submission) has been improved without the claim of fast kinetics. (2) We have revised the discussion about Supplementary Figure 5, where the description about capacitive response has been removed. (3) We also revised the discussion of Supplementary Figure 9 without the claim and/or description about fast kinetics and capacitive behavior.

Question-6: Line 99, 100, and 110: The use of charge/discharge to describe oxidation/reduction of PTO is confusing as the opposite reaction corresponds to charge and discharge in the battery cell.

Response: Thanks for your suggestions. The related discussion has been revised as: “the pristine, fully-reduced and fully-recovered (that is oxidized discharge product to the pristine state by electrochemical process) electrodes.....”.

Question-7: Line 125: The statement that the coulombic efficiency is close to 100 % is clearly incorrect (as seen in figure 2b)

Response: Thanks for kind suggestions. It has been revised to “with the coulombic efficiency of 95% at a low current density of 0.5 mA cm^{-2} ”.

Question-8: Line 192-193: It is unclear what the authors mean by “Herein, the PTO// MnO_2 @GF proton cell shows a distinct characteristic in absence of desolvation process of H_3O^+ , thus might be

able to work well at low temperature”.

Response: Many thanks for your question. We have revised the related description (please see the revised manuscript, page 7, where sentences highlighted by yellow background).

Question-9: Line 260: In addition to the areal mass loadings, please provide the electrode area used in the experiments.

Response: Thanks for your kind suggestions. The electrode area (0.6 cm²) of PTO electrode has been given in the experimental section of revised manuscript.

Question-10: A general improvement of the language is recommended.

Response: Thank you for your kind suggestions. We have tried our best to check manuscript carefully, and the language has been polished for better description of the research results and understanding of readers. If possible, we will seek help from professional agencies for language improvement of this paper for the final publication.

Response to Reviewer #3

Overall Comment: In present work, Guo et al reported a rocking-chair-type proton battery based on an organic PTO anode and MnO_2 @graphite felt anode, which works upon quinone/hydroquinone redox reaction on anode and $\text{MnO}_2/\text{Mn}^{2+}$ conversion reaction on cathode. In consideration of electrode materials exploration for proton batteries, the intercalation-type inorganic electrode materials exhibit limited performance due to the large ion radii of hydronium ions, the existing form of proton in aqueous solution. In this work, it is a very ingenious breakthrough to utilize organic/inorganic battery system based on enolization and conversion reactions rather than intercalation mechanism, thus avoiding the inherent slow kinetics and inspiring the wide choices of hydronium ion host. Such proton batteries deliver high energy density and high power density comparable to supercapacitors, as well as good cycle stability. The fabricated flexible configuration also shows good performance with outstanding energy density compared to the state-of-art reports. While the most interesting and exciting result is that the developed proton battery could work normally under such low-temperature as minus 70 degrees, which performs as a powerful member for low-temperature batteries, and might bring more inspiring research on development of low-temperature power sources. Overall, this work is interesting and important, which can be recommended for the publication in *Nature Communications* after minor revision.

Response: Thank you very much for your kindly reviewing our manuscript and giving us a positive comment. We would like to answer questions separately and revise the manuscript according to your suggestions. All the revisions according to your questions/suggestions have been highlighted by green background.

The minor concerns are summarized as following:

Question-1: The theoretical specific capacity of PTO material should be calculated as about 400 mAh/g based on four carbonyl groups per PTO molecular. While the delivered specific capacity of PTO in this work is only 208 mAh/g. Why does PTO exhibit only a half of the theoretical capacity here?

Response: Many thanks for your good question. As correctly pointed out by you, the discharge capacity of PTO electrode in three-electrode test or full battery test at a low current density is only 208 mAh g^{-1} , about half of the theoretical capacity ($\sim 400 \text{ mAh g}^{-1}$) calculated based on $4e/4H$ reaction. Actually, only about 50% carbonyl groups are involved in charge storage, indicating a $2e/2H$ reaction, which might result from that the initial H_3O^+ combination on the carbonyl groups leads to

the large steric hindrance for the further H_3O^+ combination on adjacent carbonyl groups (please also see our response to reviewer # 2).

Question-2: Both traditional and flexible configurations of the PTO-based proton batteries have been fabricated in this work. The mass loading of active material on anode (i.e., PTO) for these two configurations should be given for better understanding.

Response: Thank you for your kind suggestions. The mass loading of active material on PTO-based anode for traditional and flexible configurations are about 4 mg cm^2 and 6 mg cm^2 , respectively, which has been given in *Methods* section in the revised manuscript, page 10, highlighted with green background.

Question-3: Although such battery exhibits some inherent advantages, its disadvantages, i.e., low electronic conductivity of organic electrode, i.e., PTO, should also be mentioned in this paper.

Response: Thank you for your good suggestion. In the discussion section, we have added the discussion about the challenges of PTO electrode. Please see the revised manuscript, page 9, where sentences highlighted by yellow background).

Question-4: In Figure 2, the authors show a lot of ex-situ characterization results of $\text{MnO}_2@\text{GF}$ cathode, e.g. ex-situ XPS spectra and XRD patterns. However, the test conditions of corresponding electrochemical measurements (such as applied currents) are not given. Please clarify this point in main-text or experimental section.

Response: Thank you for your good suggestion. The test conditions of electrochemical measurements have been added into the caption of **Figure 2** in the revised manuscript, page 17, highlighted by green underground.

Question-5: When discussing the energy and power densities, the mass which calculation based on should be clearly defined, e.g., on both electrodes or not? This should be made clear when comparing to other reports, like in supplementary Table 1.

Response: Thank you for your good suggestion. The energy and power densities of our battery were calculated based on total mass of active materials (i.e. mass of PTO in anode and the mass of deposited (or dissolved) MnO_2 during charge/discharge cycles), which has been mentioned in the revised manuscript, page 6, highlighted by green background. Please also see the note of **Supplementary Table 1** of the revised supplementary information.

Question-6: The optical pictures of electrolyte under -40 and -70 degrees were given in Figure 4a. The real time temperature when the pictures were taken should be given clearly for precisely reflecting the actual state of electrolyte.

Response: Thank you for your good question. The picture of electrolyte under low temperature (i.e. -40 °C and -70 °C) in **Figure 4a** were rapidly taken in air at room temperature (25°C) after being frozen under corresponding low temperature for several hours. The explanation has been added into *Methods* section in revised manuscript, page 10, highlighted by green underground.

Reviewers' comments:

Reviewer #1 (Remarks to the Author):

I am satisfied with authors' response. The MS is now in a good shape be accepted in Nature Commun.

Reviewer #2 (Remarks to the Author):

In the revised report the Authors have addressed all questions raised by this reviewer and made corresponding changes to the manuscript. However, the low-temperature behavior is still confusing and the added discussion and evaluated diffusion coefficients rather add to the confusion than provide solid support for the observed low-temperature behavior. Without data that actually supports and explain the observed low-temperature behavior this reviewer is reluctant to recommend publication of this work.

My criticism is as follows: The cycling data presented in figure 3b, figure 4b and figure 4c suggest IR-drops on the orders of 100 mV, 120 mV and 200 mV at room temperature, at -40 Centigrade and at -70 Centigrade, respectively. (I used the 0.16mA/cm²-data to evaluate the IR-drop). Hence, going from a liquid electrolyte to a frozen electrolyte the resistance is increased by a factor of 1.2 and further increased at -70 Centigrade by a factor of 2. At the same time the diffusion coefficient is reduced by 7 orders of magnitude, from $7.12 \cdot 10^{-6}$ cm²/s at room temperature to $3.15 \cdot 10^{-13}$ cm²/s in frozen electrolyte. This effect alone should induce a dramatic increase of the cell resistance. Why is this not observed? In addition to the mass transport the two cell reactions must involve significant reorganization energy that should give activation barriers, contributing to an increased resistance at lower temperatures. Also this effect should, in this reviewer's opinion, give significantly larger effects on the kinetics compared to what is observed. As the low temperature observations are spectacular a thorough investigation on how the battery can operate in frozen electrolyte with close to no effect from mass transport must be provided. In order to make the low-temperature data convincing evaluation of the activation barriers involved in the redox reactions could also be evaluated.

Reviewer #3 (Remarks to the Author):

The manuscript is significantly improved. I recommend its publication.

Response to Reviewer#1

Comment from Reviewer #1: I am satisfied with authors' response. The MS is now in a good shape be accepted in Nature Commun.

Response: We really appreciate your kindly comment and recommending our manuscript for publication in *Nature Communications*.

Response to Reviewer#2

Comment from Reviewer #2: In the revised report the Authors have addressed all questions raised by this reviewer and made corresponding changes to the manuscript. However, the low-temperature behavior is still confusing and the added discussion and evaluated diffusion coefficients rather add to the confusion than provide solid support for the observed low-temperature behavior. Without data that actually supports and explain the observed low-temperature behavior this reviewer is reluctant to recommend publication of this work.

My criticism is as follows: The cycling data presented in figure 3b, figure 4b and figure 4c suggest IR-drops on the orders of 100 mV, 120 mV and 200 mV at room temperature, at -40 Centigrade and at -70 Centigrade, respectively. (I used the $0.16\text{mA}/\text{cm}^2$ -data to evaluate the IR-drop). Hence, going from a liquid electrolyte to a frozen electrolyte the resistance is increased by a factor of 1.2 and further increased at -70 Centigrade by a factor of 2. At the same time the diffusion coefficient is reduced by 7 orders of magnitude, from $7.12 \times 10^{-6} \text{ cm}^2/\text{s}$ at room temperature to $3.15 \times 10^{-13} \text{ cm}^2/\text{s}$ in frozen electrolyte. This effect alone should induce a dramatic increase of the cell resistance. Why is this not observed? In addition to the mass transport the two cell reactions must involve significant reorganization energy that should give activation barriers, contributing to an increased resistance at lower temperatures. Also this effect should, in this reviewer's opinion, give significantly larger effects on the kinetics compared to what is observed. As the low temperature observations are spectacular a thorough investigation on how the battery can operate in frozen electrolyte with close to no effect from mass transport must be provided. In order to make the low-temperature data convincing evaluation of the activation barriers involved in the redox reactions could also be evaluated.

Response: We really appreciate your comment. However, you misunderstand our previous response, which arises from our poor description. In fact, the conductivity or the related diffusion coefficient of our electrolyte is only reduced by ~ 2 orders of magnitude from room temperature (25 °C) to -70 °C. Here we would like to response to your comment as follows:

- 1) The Mn^{2+} diffusion coefficient (i.e. $3.15 \times 10^{-13} \text{ cm}^2 \text{ S}^{-1}$ given in our previous response and/or the related revised manuscript) is calculated by us with Nernst-Einstein equation, which is based on a reported conductivity of a frozen 1 M MnSO_4 solution at -10 °C (see ref. 46, *Angew. Chem. Int.*

Ed. 2019, 58, 12569-12573). This value is only used to demonstrate that Mn^{2+} can even diffuse in a frozen solution (Sorry for that we did not describe this point clearly). Herein, it should be noted that the electrolyte in our battery is (2 M $MnSO_4$ + 2 M H_2SO_4), rather than 1 M $MnSO_4$ solution. That is to say, such value for 1 M $MnSO_4$ solution cannot be directly used to evaluate the low temperature performance of our battery using the frozen (2 M H_2SO_4 + 2 M $MnSO_4$) electrolyte. The conductivity and the related Mn^{2+} diffusion coefficient of the frozen (2 M $MnSO_4$ + 2 M H_2SO_4) solution should be quite different from that of the frozen $MnSO_4$ solution (see next point).

- 2) Electrochemical impedance spectroscopy (EIS) measurements were carried out to clarify the conductivity of the frozen (2 M $MnSO_4$ + 2 M H_2SO_4) electrolyte and 2 M $MnSO_4$ electrolyte at $-70\text{ }^\circ\text{C}$ (see **Figure answer 1**).

Figure answer 1. Nyquist plot of (a) 2 mol L^{-1} $MnSO_4$ + 2 mol L^{-1} H_2SO_4 and (b) 2 mol L^{-1} $MnSO_4$ solutions at $-70\text{ }^\circ\text{C}$. (The EIS measurement details are given in method section in the main text.)

According to **Figure answer 1**, the (2 M $MnSO_4$ + 2 M H_2SO_4) electrolyte and 2 M $MnSO_4$ electrolyte show the conductivities of $2.66 \times 10^{-3}\text{ S cm}^{-1}$ and $7.962 \times 10^{-7}\text{ S cm}^{-1}$, respectively, at $-70\text{ }^\circ\text{C}$. Obviously, the conductivity of the frozen (2 M $MnSO_4$ + 2 M H_2SO_4) electrolyte is much higher than that of the frozen 2 M $MnSO_4$ electrolyte at $-70\text{ }^\circ\text{C}$ (This point will be further discussed later). Furthermore, EIS measurement was also carried out to evaluate the conductivity of the (2 M $MnSO_4$ + 2 M H_2SO_4) electrolyte at room temperature ($25\text{ }^\circ\text{C}$) (see **Figure answer 2**). According to **Figure answer 2**, the conductivity of (2 M $MnSO_4$ + 2 M H_2SO_4) electrolyte at room

temperature (25 °C) is derived as $5.31 \times 10^{-1} \text{ S cm}^{-1}$, which is only two orders of magnitude higher than the conductivity at -70 °C ($2.66 \times 10^{-3} \text{ S cm}^{-1}$).

Figure answer 2. Nyquist plot of 2 mol L⁻¹ MnSO₄ + 2 mol L⁻¹ H₂SO₄ solution at room temperature (25 °C). (The EIS measurement details are given in method section in the main text.)

The diffusion coefficient could be estimated through Nernst-Einstein equation (*Equation 1*).

$$\sigma = \frac{N_A}{k_B T} \sum_i^n c_i q_i^2 D_i \quad (1)$$

Where σ is the ion conductivity, N_A the Avogadro number, e the elementary charge, K_B the Boltzmann constant, T the absolute temperature of the system, and c_i , q_i , D_i , respectively, the concentration, charge, and diffusion coefficient of ions of species i (cations or anions). For convenience, we consider $D_+ \approx D_-$ (just for estimation). As a result, the diffusion coefficient of Mn^{2+} in a frozen 2 M MnSO₄ electrolyte (at -70 °C) is calculated as $9.052 \times 10^{-15} \text{ cm}^2 \text{ S}^{-1}$. For the (2 M MnSO₄ + 2 M H₂SO₄) electrolyte (with the consideration of $D_{\text{H}^+} \approx D_{\text{Mn}^{2+}} \approx D_{\text{SO}_4^{2-}}$), the diffusion coefficient of Mn^{2+} at -70 °C is calculated as $1.728 \times 10^{-11} \text{ cm}^2 \text{ S}^{-1}$. With the same way, the diffusion coefficient of Mn^{2+} in (2 M MnSO₄ + 2 M H₂SO₄) electrolyte at 25 °C can be calculated as $3.450 \times 10^{-9} \text{ cm}^2 \text{ S}^{-1}$, which is about two orders of magnitude higher than that at -70 °C. That is to say, the diffusion coefficient of Mn^{2+} is only reduced by 2 orders of magnitude from $3.450 \times 10^{-9} \text{ cm}^2 \text{ S}^{-1}$ at room temperature to $1.728 \times 10^{-11} \text{ cm}^2 \text{ S}^{-1}$ at -70 °C.

- 3) As mentioned above, the conductivity and the related ion diffusion coefficient of the frozen (2 M MnSO₄ + 2 M H₂SO₄) solution is much higher than that of the frozen 1 M MnSO₄ solution. One reason for this phenomenon is that the presence of proton and the higher concentration reduces

the freezing point greatly. To clarify this point, differential scanning calorimetry (DSC) measurement was employed to analyze the freezing points of (2 M MnSO₄ + 2 M H₂SO₄) and (2 M MnSO₄) solutions, which are summarized in **Figure answer 3**. As shown in **Figure answer 3**, the (2 M MnSO₄ + 2 M H₂SO₄) solution exhibits the freezing point of approximately -41.6 °C, which is much lower than that of MnSO₄ solution (-11.6 °C).

Figure answer 3. Differential scanning calorimetry (DSC) measurement of (a) 2 mol L⁻¹ MnSO₄ + 2 mol L⁻¹ H₂SO₄ and (b) 2 mol L⁻¹ MnSO₄ solutions. (The DSC measurement details are given in method section in the main text.)

PS:

According to your comment, **Figures answer 1-3** and the related discussions have been given in the revised supplementary information (see **Supplementary Figures 19, 20, 21** and the related discussions about Mn²⁺ diffusion coefficients). The conductivities of the electrolyte (2 M MnSO₄ + 2 M H₂SO₄) obtained from EIS data and the estimated Mn²⁺ diffusion coefficients at room temperature and at -70 °C have been given in the revised manuscript (see page 8, where sentences highlighted by yellow background), which can be used to explain the limited IR-drop even at ultra-low temperature.

By the way, according the publication policy of *Nature Communications*, we also would like to publish “the response letters to reviewers” online, which is coupled with our manuscript (after it is accepted). This will help popular readers to further understand the low temperature performance, since your comment is very good and important.

Finally, it should be noted that the ion diffusion in a frozen electrolyte is still a new and complex topic for battery research, which needs further investigations in the future. We have mentioned this point in the revised manuscript.

Response to Reviewer#3

Comment from Reviewer #3: The manuscript is significantly improved. I recommend its publication.

Response: Thank you very much for your kindly comment and recommending our manuscript for publication in *Nature Communications*.

Reviewers' comments:

Reviewer #2 (Remarks to the Author):

I appreciate the clarification and thorough response on my remaining concern. In my opinion there are, however, critical questions regarding the low temperature behavior that should be addressed prior to publication, but I will (of course) leave the final decision to the editor. Two questions still need to be clarified 1) how is it possible, at all, to operate the presented battery at freezing conditions and 2) why are the effect of lowering the temperature below the electrolyte freezing point so small (close to negligible)?

The first question relate to that Mn^{2+} diffusion should be frozen out and any electron transfer event are expected to become energetically unfavorable as the electrolyte can no longer rearrange to counter balance the new electronic configuration. Hence, the electron transfer reaction must occur without solvent rearrangement. In water solution the outer reorganization energy (the energy required to transfer an electron without rearranging the nuclei) is usually around 1 eV. The effect of freezing out electron transfer reactions is well documented in the electron transfer literature. It is possible that the high salt/acid concentration lessen this effect and that electron transfers in this particular electrolyte is possible, but perhaps not very likely based on current knowledge. The results provided in ref 46 indeed suggest that Mn^{2+} diffusion is possible in frozen media and hence provide support that idea. So, there might be ways to account for a working battery with frozen electrolyte although consensus in the scientific literature (known to me) point in another direction.

The second question is more critical and I pointed this out in my last review report. Why are the effects of freezing the electrolyte so small? You state that the conductivity is reduced by 2 orders of magnitude (more on this below). Why does the cell resistance decrease only by a factor of 1.2 and 2 at -40 centigrades and -70 centigrades, respectively? From the decrease in electrolyte conductivity alone a much larger increase in resistance is expected. And, in addition to electrolyte resistance the charge transfer resistance should give additional contributions to the cell resistance due (partially) to the effects above.

As I stated in my first response it is likely that proton diffusion dominate the ion transport in frozen media. When comparing the resistance in 1M $MnSO_4$ (from literature) with 2M $MnSO_4 + 2M H_2SO_4$ (present work) the authors state that the difference in conductivity seen in frozen solution between these electrolytes is due to a decreased freezing point in the acidic solution. An alternative explanation, and a more likely explanation in my opinion, is that the increased conductivity in the acidic electrolyte is due to the diffusion of protons. The assumption that the diffusion coefficient for H^+ , Mn^{2+} and SO_4^{2-} are equal is a gross over-simplification known to be wrong, in particular in frozen solution. Hence, even though the electrolyte resistance is reduced by two orders of magnitude it is likely that the Mn^{2+} resistivity is much higher. Since Mn^{2+} diffusion is required, based on the cell reactions, an even larger effect, than the two orders of magnitude drop in electrolyte resistance would suggest, is expected.

Response to Reviewer #2

[Remarks to the Author: I appreciate the clarification and thorough response on my remaining concern. In my opinion there are, however, critical questions regarding the low temperature behavior that should be addressed prior to publication, but I will (of course) leave the final decision to the editor. Two questions still need to be clarified 1) how is it possible, at all, to operate the presented battery at freezing conditions and 2) why are the effect of lowering the temperature below the electrolyte freezing point so small (close to negligible)?

The first question relate to that Mn^{2+} diffusion should be frozen out and any electron transfer event are expected to become energetically unfavorable as the electrolyte can no longer rearrange to counter balance the new electronic configuration. Hence, the electron transfer reaction must occur without solvent rearrangement. In water solution the outer reorganization energy (the energy required to transfer an electron without rearranging the nuclei) is usually around 1 eV. The effect of freezing out electron transfer reactions is well documented in the electron transfer literature. It is possible that the high salt/acid concentration lessen this effect and that electron transfers in this particular electrolyte is possible, but perhaps not very likely based on current knowledge. The results provided in ref 46 indeed suggest that Mn^{2+} diffusion is possible in frozen media and hence provide support that idea. So, there might be ways to account for a working battery with frozen electrolyte although consensus in the scientific literature (known to me) point in another direction.

The second question is more critical and I pointed this out in my last review report. Why are the effects of freezing the electrolyte so small? You state that the conductivity is reduced by 2 orders of magnitude (more on this below). Why does the cell resistance decrease only by a factor of 1.2 and 2 at -40 centigrades and -70 centigrades, respectively? From the decrease in electrolyte conductivity alone a much larger increase in resistance is expected. And, in addition to electrolyte resistance the charge transfer resistance should give additional contributions to the cell resistance due (partially) to the effects above.

As I stated in my first response it is likely that proton diffusion dominate the ion transport in frozen media. When comparing the resistance in 1M MnSO_4 (from literature) with 2M $\text{MnSO}_4 + 2\text{M H}_2\text{SO}_4$ (present work) the authors state that the difference in conductivity seen in frozen solution between these electrolytes is due to a decreased freezing point in the acidic solution. An alternative explanation, and a more likely explanation in my opinion, is that the increased conductivity in the acidic electrolyte is due to the diffusion of protons. The assumption that the diffusion coefficient for H^+ , Mn^{2+} and SO_4^{2-} are equal is a gross over-simplification known to be wrong, in particular in frozen solution. Hence, even though the electrolyte resistance is reduced by two orders of magnitude it is likely that the Mn^{2+} resistivity is much higher. Since Mn^{2+} diffusion is required, based on the cell reactions, an even larger effect, than the two orders of magnitude drop in electrolyte resistance would suggest, is expected.]

Response: Many thanks for your good suggestions and questions, which is really important. It seems that in your opinion, our results (i.e., the achieved performance at a super-low temperature) are doubtful. Thus, here we would like to answer as follows:

(1) We state that all the experiment data are credible and repeatable. We have repeated our experiment of low temperature test on Jan. 12th 2020, again. A short video about this experiment has

been prepared by Prof. Y. W, Z. G, L. Y and J. H, in order to clarify the details of our experiment. We would like to publish this video and related introduction (see page 4 of this letter), in parallel with our manuscript and our response to reviewers. After its publication, any reader and/or you can directly contact us for discussion about this experiment or get the raw data that is shown in this video. Now, if you really think that our data are interesting and important, recommendation for publication might be the best solution, through which the readers can discuss about it, repeat it, continue it and even argue about it. Especially, I sincerely hope you could understand that this work has been reviewed for 7 months. If possible, we even hope to co-operate with you to further clarify this interesting results in future research.

(2) It should be noted that the estimated Mn^{2+} diffusion coefficient is only used to demonstrate that Mn^{2+} can move in a frozen electrolyte. However, it cannot be used to verify our experiment data. It is almost impossible to calculate an exact Mn^{2+} diffusion coefficient in such a complex electrolyte ($H^+ + Mn^{2+} + SO_4^{2-}$) coupled with a complex electrode reaction in cathode, especially in a frozen electrolyte at super-low temperature. Herein, we further explain our previous response as follows:

Firstly, according to the reported conductivity of a frozen 1 M $MnSO_4$ solution at (ref. 46), we estimated a Mn^{2+} diffusion coefficient (i.e. $3.15 \times 10^{-13} \text{ cm}^2 \text{ S}^{-1}$) using Nernst-Einstein equation with a consideration of $D_+ \approx D_-$. This value is only used to demonstrate that Mn^{2+} can move in a frozen electrolyte. [PS: this value may be lower than the real diffusion coefficient of Mn^{2+} in the (2 M H_2SO_4 + 2 M $MnSO_4$) electrolyte because of the presence of proton and/or proton-induced lower freezing point of (2 M H_2SO_4 + 2 M $MnSO_4$) solution.]

Secondly, according to the achieved EIS data, we also estimated a Mn^{2+} diffusion coefficient in the (2 M H_2SO_4 + 2 M $MnSO_4$) electrolyte at room temperature ($3.450 \times 10^{-9} \text{ cm}^2 \text{ S}^{-1}$) and the frozen (2 M H_2SO_4 + 2 M $MnSO_4$) electrolyte at $-70 \text{ }^\circ\text{C}$ ($1.728 \times 10^{-11} \text{ cm}^2 \text{ S}^{-1}$) using Nernst-Einstein equation with a consideration of $D_{H^+} \approx D_{Mn^{2+}} \approx D_{SO_4^{2-}}$. [PS: We agree with you that the estimated values should be higher than the real diffusion coefficient of Mn^{2+} in the electrolyte, because proton generally exhibits much higher diffusion rate than Mn^{2+} and SO_4^{2-} . However, for present case, it is very difficult to give an exact value of Mn^{2+} diffusion coefficient, owing to the mixture of proton, Mn^{2+} and SO_4^{2-} in one solution.]

Thirdly, as correctly pointed out by you, the redox reaction at the cathode side requires diffusion of both water molecules (and hydronium ions) and Mn^{2+} ions, which involves both the deposition/dissolution of MnO_2 and the generation/consumption of hydronium ions. [PS: That is to say, it is really difficult to distinguish the diffusion of Mn^{2+} from that of proton during the electrode reaction. Is there a synergy (i.e., proton diffusion facilitates Mn^{2+} diffusion)? This question needs further research in the future.]

In summary, we emphasize again that these estimated Mn^{2+} diffusion coefficients are only used to demonstrate that Mn^{2+} can move in a frozen electrolyte. Certainly, we also agree with that the assumption of $D_{H^+} \approx D_{Mn^{2+}} \approx D_{SO_4^{2-}}$ is a gross over-simplification. Therefore, we have removed the discussion about Mn^{2+} diffusion coefficients from the main-text of revised manuscript, in order to prevent misunderstanding. At the same time, we would like publish all of our responses to reviewers (that is permitted by Nature Communications), through which all the readers can see our discussion

about the Mn^{2+} diffusion rate. In addition, it is emphasized in the revised manuscript that “*it is still very difficult to calculate an exact diffusion coefficient of Mn^{2+} in the hybrid electrolyte containing proton, Mn^{2+} and SO_4^{2-} , especially at the frozen situation. Ion diffusion in a frozen electrolyte should be a new and complex topic, which needs further investigations in the future.*”(see page 9). We sincerely hope that you can recommend for publication of this work at the present form, through which you and/or the readers can directly discuss about it, repeat it, continue it and even argue about it.

Video introduction

This video includes 6 parts (A-F):

A: Fabrication of the PTO//MnO₂@GF full battery. The glass battery container was used at room temperature for demonstration. For charge/discharge test at low temperature (-70°C), the polytetrafluoroethylene (PTFE) container was used, in view of that the glass container would frost crack at such low temperature.

B: Prior to the charge/discharge test at -70°C, the battery was first cycled at -65°C for 5 cycles, as we mentioned in methods section in the main-text. And all the charge/discharge curves at various current densities were obtained using one battery, tested by Landt electrochemical station.

The charge-discharge curve of the PTO//MnO₂@GF full battery at -70°C tested at a current density of 0.4 mA cm⁻², recorded at various charge/discharge states (1-13):

- (1) Charge to 0.75 V, with a charge capacity of 5 mAh g⁻¹;
- (2) Charge to 0.91 V, with a charge capacity of 16 mAh g⁻¹;
- (3) Charge to 0.94 V, with a charge capacity of 33 mAh g⁻¹;
- (4) Charge to 1.02 V, with a charge capacity of 57 mAh g⁻¹;
- (5) Charge to 1.05 V, with a charge capacity of 75 mAh g⁻¹;
- (6) Charge to 1.11 V, with a charge capacity of 97 mAh g⁻¹;
- (7) Discharge to 0.94 V, with a discharge capacity of 10 mAh g⁻¹;
- (8) Discharge to 0.91 V, with a discharge capacity of 22 mAh g⁻¹;
- (9) Discharge to 0.86 V, with a discharge capacity of 44 mAh g⁻¹;
- (10) Discharge to 0.81 V, with a discharge capacity of 53 mAh g⁻¹;
- (11) Discharge to 0.74 V, with a discharge capacity of 86 mAh g⁻¹;
- (12) Discharge to 0.69 V, with a discharge capacity of 94 mAh g⁻¹;
- (13) Discharge to 0.3 V, with a discharge capacity of 106 mAh g⁻¹.

C: Charge-discharge curve of the full battery at -70°C tested at a current density of 0.16 mA cm⁻².

D: Charge-discharge curve of the full battery at -70°C tested at a current density of 0.8 mA cm⁻².

E: Charge-discharge curve of the full battery at -70°C tested at a current density of 2 mA cm⁻².

F: Demonstration of the frozen battery at -70°C after the above charge/discharge tests. PS: the whole battery for demonstration was rest for 10 minutes after the freezer was shut down. Then, the battery was taken out from the freezer for demonstration.

REVIEWERS' COMMENTS:

Reviewer #2 (Remarks to the Author):

Based on my knowledge the presented battery should not work at all in frozen electrolyte and I consider the fact that it does work to be highly unexpected. When you come across something truly spectacular and unexpected it is, in my opinion, our responsibility as researchers to find out why nature behave in such unexpected manner. Whether this should be done prior to or after publication of the spectacular result depends on the risk that the publisher is willing to take and is hence, in my opinion, an editorial decision to make. I have no doubt in that you report what you observe and the risk, in this case, is that the observed behavior is due to something more trivial. Excluding all uncertainties in experimental research is impossible and I therefor favor demand for coherent models accounting for the experimental observations reported. In the presented work the low-temperature behavior is in conflict with conductance data presented in the manuscript as well as with current knowledge regarding electron transfer and ion diffusion in frozen media, all of which suggest that the cell resistance should increase by order(s) of magnitudes if the electrolyte is frozen. I do, however, appreciate the limited resources available to each individual research group and the possibility that other research groups may be better equipped to provide a coherent model for the data presented in the present study. In order not to delay publication any further I therefor recommend publishing the manuscript in its present form.

Response to Reviewer # 2

Overall Comment: Based on my knowledge the presented battery should not work at all in frozen electrolyte and I consider the fact that it does work to be highly unexpected. When you come across something truly spectacular and unexpected it is, in my opinion, our responsibility as researchers to find out why nature behave in such unexpected manner. Whether this should be done prior to or after publication of the spectacular result depends on the risk that the publisher is willing to take and is hence, in my opinion, an editorial decision to make. I have no doubt in that you report what you observe and the risk, in this case, is that the observed behavior is due to something more trivial. Excluding all uncertainties in experimental research is impossible and I therefor favor demand for coherent models accounting for the experimental observations reported. In the presented work the low-temperature behavior is in conflict with conductance data presented in the manuscript as well as with current

knowledge regarding electron transfer and ion diffusion in frozen media, all of which suggest that the cell resistance should increase by order(s) of magnitudes if the electrolyte is frozen. I do, however, appreciate the limited resources available to each individual research group and the possibility that other research groups may be better equipped to provide a coherent model for the data presented in the present study. In order not to delay publication any further I therefor recommend publishing the manuscript in its present form.

Response: Many thanks for kindly reviewing our manuscript and giving a final recommendation of “*publishing the manuscript in its present form*”. All the responses to reviewers will be published in parallel with our manuscript, which might evoke popular attention on our work and your comments. We also hope that this interesting phenomenon can be further clarified by our group and/or other research groups in the near future.